# Longer and more frequent marine heatwaves over the past century

Eric C.J. Oliver [1,2,3], Markus G. Donat [4,5], Michael T. Burrows[6], Pippa J. Moore[7], Dan A. Smale [8,9], Lisa V. Alexander[4,5], Jessica A. Benthuysen[10], Ming Feng [11], Alex Sen Gupta [4,5], Alistair J. Hobday[12], Neil J. Holbrook [2,13], Sarah E. Perkins-Kirkpatrick[4,5], Hillary A. Scannell[14,15], Sandra C. Straub [9] & Thomas Wernberg [9]

Heatwaves are important climatic extremes in atmospheric and oceanic systems that can have devastating and long-term impacts on ecosystems, with subsequent socioeconomic consequences. Recent prominent marine heatwaves have attracted considerable scientific and public interest. Despite this, a comprehensive assessment of how these ocean temperature extremes have been changing globally is missing. Using a range of ocean temperature data including global records of daily satellite observations, daily in situ measurements and gridded monthly in situ-based data sets, we identify significant increases in marine heatwaves over the past century. We find that from 1925 to 2016, global average marine heatwave frequency and duration increased by 34% and 17%, respectively, resulting in a 54% increase in annual marine heatwave days globally. Importantly, these trends can largely be explained by increases in mean ocean temperatures, suggesting that we can expect further increases in marine heatwave days under continued global warming.

[1] Department of Oceanography, Dalhousie University, 1355 Oxford Street, Halifax, NS B3H 4R2, Canada. [2] Institute for Marine and Antarctic Studies, University of Tasmania, 20 Castray Esplanade, Battery Point, Private Bag 129, Hobart, TAS 7001, Australia. [3] Australian Research Council Centre of Excellence for Climate System Science, University of Tasmania, Private Bag 129, Hobart, TAS 7001, Australia. [4] Climate Change Research Centre, University of New South Wales, Gate 11 Botany Street, Library Walk, Level 4, Matthews Building, Sydney, NSW 2052, Australia. [5] Australian Research Council Centre of Excellence for Climate System Science, University of New South Wales, Gate 11 Botany Street, Library Walk, Level 4, Matthews Building, Sydney, NSW 2052, Australia. [6] Scottish Association for Marine Science, Scottish Marine Institute, Oban, Argyll, PA37 1QA Scotland, UK. [7] Institute of Biological, Environmental and Rural Sciences, Aberystwyth University, Aberystwyth, SY23 3DA, UK. [8] Marine Biological Association of the United Kingdom, The Laboratory, Citadel Hill, Plymouth, PL1 2PB, UK. [9] UWA Oceans Institute and School of Biological Sciences, The University of Western Australia, Crawley, WA 6009, Australia. [10] Australian Institute of Marine Science, PMB 3, Townsville MC, QLD 4810, Australia. [11] CSIRO Oceans and Atmosphere, Crawley, 6009 WA, Australia. [12] CSIRO Oceans and Atmosphere, Hobart, TAS 7000, Australia. [13] Australian Research Council Centre of Excellence for Climate Extremes, University of Tasmania, Private Bag 129, Hobart, TAS 7001, Australia. [14] School of Oceanography, University of Washington, Seattle, 98105 WA, USA. [15] NOAA Pacific Marine Environmental Laboratory, Seattle, 98115 WA, USA. Correspondence and requests for materials should be addressed to E.C.J.O. (email: eric.oliver@dal.ca)

Several prominent marine heatwaves (MHWs)—prolonged periods of anomalously high sea surface temperatures[1]—have had severe impacts on marine ecosystems in recent years. Notable events occurred in the northern Mediterranean Sea in 2003[2,3], along the Western Australian coast in 2011[4], the northwest Atlantic in 2012[5], the northeast Pacific over 2013–2015[6,7], off southeastern Australia in 2015/16[8] and across northern Australia in 2016[9]. These events resulted in substantial ecological and economic impacts, including sustained loss of kelp forests[10], coral bleaching[11], reduced surface chlorophyll levels due to increased surface layer stratification[6], mass mortality of marine invertebrates due to heat stress[8,12], rapid long-distance species' range shifts and associated reshaping of community structure[8,10,13], fishery closures or quota changes[8,13,14] and even intensified economic tensions between nations[15]. Such impacts demonstrate the damaging consequences of MHWs and their influence on the structure and sustainability of marine communities and ecosystems[4,8,10,12,13,16–19]. Given the expected intensification in extreme temperature events due to anthropogenic climate change[20] and the potential for profound ecological and socioeconomic impacts, quantifying trends and patterns of MHWs is a pressing issue.

Upper ocean temperatures have warmed significantly in most regions of the world over recent decades, with anthropogenic greenhouse gas forcing very likely being the main contributor[21]. Species are often more strongly impacted by environmental extremes than by slow changes in mean conditions[22]. In a warming climate some of the most dramatic ecosystem changes have been associated with extreme heatwaves[23]. Marine heatwaves have been accompanied by large-scale shifts in marine species location, phenological changes, changes in ecosystem structure and in some cases high levels of mortality, often with socioeconomic consequences[24–27] and widespread and devastating ecological and socioeconomic consequences[28,29]. While marine heatwaves have only started to draw attention in recent years, the increasing intensity, frequency and duration of atmospheric heatwaves have been extensively documented[30]. Atmospheric heatwaves can have significant impacts on human health[31] and attribution studies have shown that these events, and atmospheric heatwaves in general, have become much more likely as a result of anthropogenic warming[32]. For example, the 2003 European heatwave caused tens of thousands of deaths[33] and was later superseded in intensity by the 2010 European heatwave, events which can be expected to increase in probability over the 2010–2050 period[34]. While centennial increases in ocean surface temperatures have been extensively reported[35,36], global trends in ocean temperature extremes remain largely unexplored.

Here, we report on local and global changes in MHW characteristics over time as recorded by satellite and in situ measurements of sea surface temperature (SST) and defined using a quantitative MHW framework, which allows for comparisons across regions and events[1]. We used three independent and complementary sources of SST data to reveal global patterns of change in MHW frequency, intensity and duration over the past century. Our analysis reveals significant centennial-scale increases in MHW properties as well as variability on interannual-to-multidecadal time scales. We find significant secular increases in both the frequency and duration of marine heatwaves, amounting globally to a 54% increase in annual marine heatwave days between 1925–1954 and 1987–2016. The data over the satellite period (1982–2016) accounts for a greater portion of this increase than in situ-based records do (going back to 1900), indicating an accelerating trend. We also find that changes in mean SST alone can account for the majority of these changes, at least over the satellite record. Finally, the El Niño-Southern Oscillation, Pacific Decadal Oscillation and Atlantic Multidecadal Oscillation each contribute to large variations in MHWs both regionally and globally.

## Results

**Marine heatwaves over the satellite record.** We applied a standardised MHW definition (see 'Methods' and ref. [1]) to global daily remotely sensed National Oceanic and Atmospheric Administration (NOAA) Optimum Interpolation (OI) SST V2 high resolution (1/4°) gridded SST data over 1982–2016[37,38]. Following this definition, MHWs occur when SSTs exceed a seasonally varying threshold, defined as the 90th percentile of SST variations based on a 30-year climatological period (1983–2012), for at least five consecutive days. At each location and for each MHW, we calculated the event duration (time between start and end dates) and maximum intensity (peak SST anomaly over the event duration). Associated annual statistics were then calculated, including the frequency of events (i.e., the number of discrete events occurring in each year), mean annual duration, maximum annual intensity and the total number of MHW days per year.

MHW frequency ranged from about one to three annual events, depending on location (Fig. 1a). The most notable exception was in the eastern tropical Pacific, where El Niño-Southern Oscillation (ENSO) events manifest as individual long-duration MHWs and resulted in less than one MHW per year on average. As a global average, MHW frequency increased significantly with a trend of 0.45 annual events per decade over 1982–2016 ($p < 0.01$; Fig. 1c, black line). This is equivalent to an average increase of 1.6 annual events, or more than five additional MHW days, by the end of the 35-year record. MHW frequency increased over 82% of the global ocean between two 17-year periods at the beginning and end of the record (1982–1998 and 2000–2016, splitting the time series in half; Fig. 1b). The largest increase occurred in the high-latitude North Atlantic Ocean (north of 50° N; an increase of 2–6 annual events). More moderate increases occurred in the subtropical portions of the North and South Atlantic Ocean, central and western portions of the North and South Pacific Ocean, and parts of the Indian Ocean (1–4 annual events). Conversely, MHW frequency decreased over parts of the Southern Ocean poleward of 50° S (particularly the Pacific sector; up to 3 fewer annual events) and parts of the eastern Pacific Ocean (as much as 2 fewer annual events).

Large spatial variations were prevalent in the intensity of MHWs (Fig. 1d). Hotspots of high intensity occurred in regions of large SST variability including the five western boundary current extension regions (+2–5 °C), the central and eastern equatorial Pacific Ocean (+1–4 °C) and eastern boundary current regions (+1–3 °C). MHW intensity between 1982–1998 and 2000–2016 increased in over 65% of the global ocean, most notably in all five western boundary current regions, where the mean warming has been considerably faster than the global average[39], and most mid-latitude ocean basins (Fig. 1e). Decreases in MHW intensity were found across much of the Tropics, most notably in the eastern tropical Pacific Ocean. The linear trend in global average MHW intensity was +0.085 °C per decade ($p < 0.01$; Fig. 1f), punctuated by large interannual variability. Interestingly, this trend was weaker than the global SST warming trend (+0.16 °C per decade, $p < 0.01$; Fig. 1l).

The typical duration of MHWs varied considerably across the global ocean (Fig. 1g). The eastern tropical Pacific, a region dominated by ENSO SST variability, had an average duration of up to 60 days, while other tropical regions were typically characterised by events of 5–10 days. Across extra-tropical regions, mean event durations were more uniform (10–15 days), with the exception of the northeast and southeast Pacific Ocean with up to 30-day mean durations. Mean MHW duration between

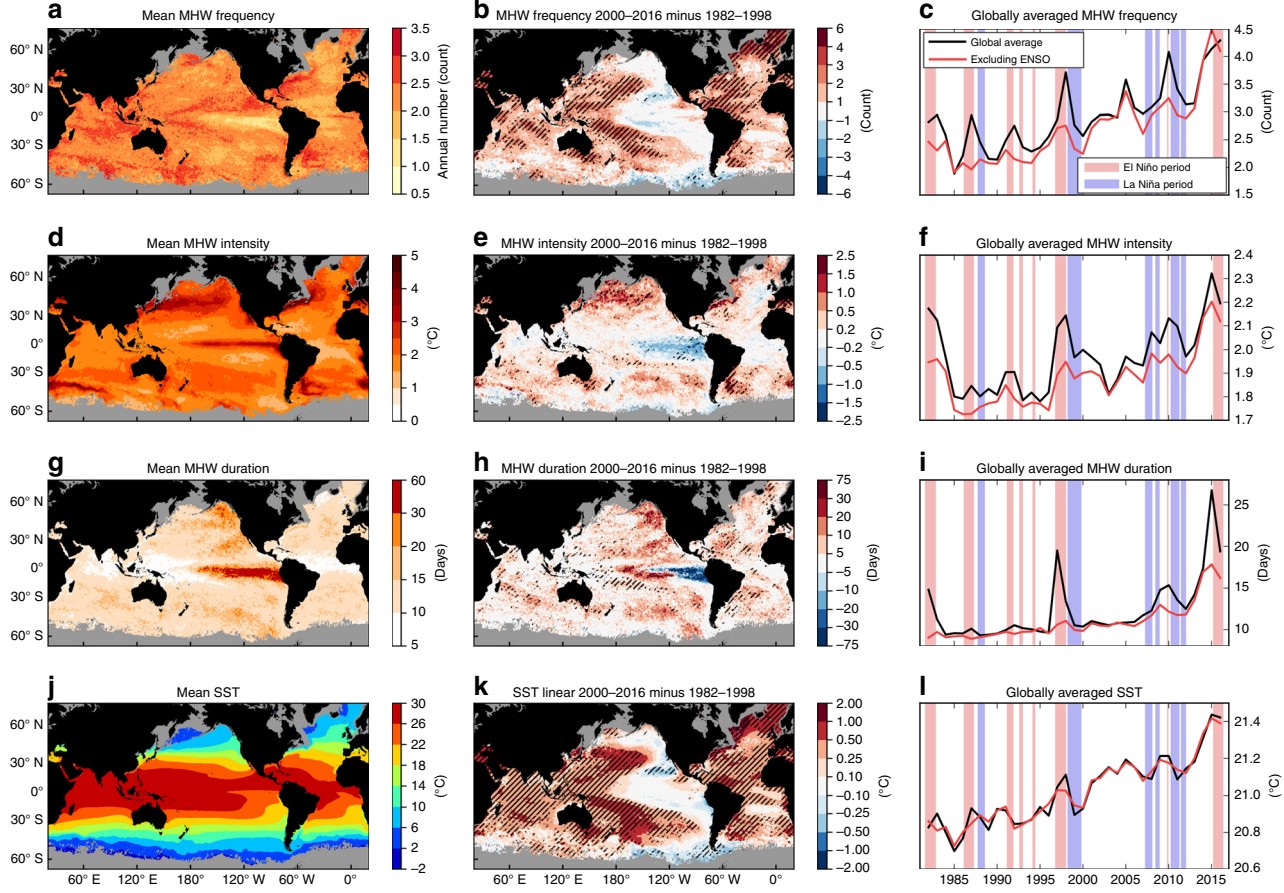

**Fig. 1** Global sea surface temperature and marine heatwave properties. **a**, **d**, **g**, **j** Mean over the 1982–2016 period, **b**, **e**, **h**, **k** difference between the 1982–1998 and 2000–2016 periods, and **c**, **f**, **i**, **l** globally averaged time series of annual mean **a–c** marine heatwave (MHW) count, **d–f** MHW intensity, **g–i** MHW duration and **j–l** SST from NOAA OI SST over 1982–2016. In **b**, **e**, **h** and **k**, hatching indicates the change is significantly different from zero at the 5% level. In **c**, **f**, **i** and **l**, the black lines show the globally averaged time series and red lines show a global average after removing the signature of ENSO. In **c**, **f**, **i** and **l**, the light red and blue shading indicate El Niño and La Niña periods, respectively, defined by periods exceeding ±1 s.d. of the MEI index for at least three consecutive months

the 1982–1998 and 2000–2016 periods increased across 84% of the global ocean, with significant increases of up to 20 days in the mid- and high-latitude regions of all ocean basis, up to 30+ days in the central tropical Pacific Ocean and northeastern Pacific Ocean, and decreases in the eastern tropical Pacific Ocean and the high latitudes of the Southern Ocean (Fig. 1h). Averaged across the global ocean, mean MHW durations have become significantly longer by 1.3 days per decade ($p < 0.01$) since 1982. The increases in frequency and duration metrics translate to 30 additional marine heatwave days per year by the end of the 35-year period ($p < 0.01$; based on a linear trend) from a baseline level of about 25 days in the 1980s (Fig. 2).

There was a clear relationship between the average frequency and duration of MHWs at each location (Fig. 1a, g). The spatial pattern of mean MHW frequency and duration was negatively correlated ($r = -0.50$), indicating that duration was long where frequency was low, and vice versa. By definition, 10% of SST days in each time series were candidates for MHW events due to the use of the 90th percentile threshold; although less than 10% were actually identified as part of MHWs due to the requirement of a duration of at least five consecutive days. Nonetheless, to maintain this roughly constant number of MHW days per year, a longer average duration implies a lower mean frequency. On the other hand, the spatial distribution of MHW intensity was largely determined by the local temperature variability. In fact, the spatial patterns of the standard deviation of annual mean SST (not

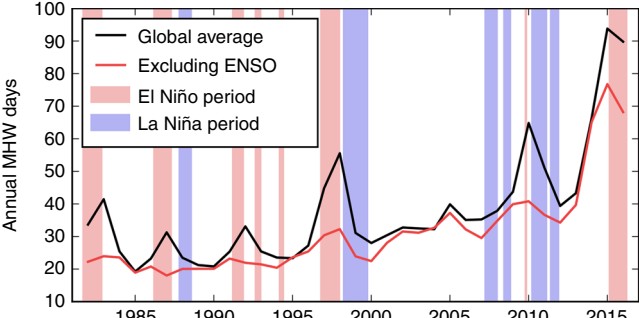

**Fig. 2** Total number of marine heatwave days globally. Globally averaged time series of total marine heatwave (MHW) days from NOAA OI SST over 1982–2016. The black line shows the globally averaged time series of total MHW days from NOAA OI SST over 1982–2016. The red line shows this metric after removing the signature of ENSO. The light red and blue shading indicate El Niño and La Niña periods, respectively, defined by periods exceeding ±1 s.d. of the MEI index for three consecutive months

shown) and mean MHW intensity (Fig. 1d) were strongly correlated ($r = 0.77$).

**The role of climate variability.** It is important when considering changes in MHWs to distinguish between the roles of secular climate change, which has in large part been attributed to

anthropogenic factors, and of transient climate variability, which is largely intrinsic. The globally averaged MHW property time series exhibit clear interannual variability (Fig. 1c, f, i, black lines). Such variability is most easily quantified by examining climate modes, and the dominant mode globally is ENSO[35]. We removed the influence of ENSO using a statistical model which predicts the linear signature of ENSO on SST at each pixel, and recalculated the MHW properties with this signature removed (see 'Methods'). After removing the effect of ENSO, the globally averaged MHW properties exhibited reduced interannual variability (Fig. 1c, f, i, red lines).

Frequency and intensity appear to relate to ENSO events such that most strong and moderate intensity El Niño events (1982/83, 1986/87/88, 1991/92, 1997/98, 2004/05, 2009/10 and 2015/16) are evident as a peak in globally averaged frequency and intensity (Fig. 1c, f). Duration appears to respond nonlinearly to ENSO so that only strong El Niño years (1982/83, 1997/98, 2015/16 and to a lesser extent 2009/10) are evident in the globally averaged duration (Fig. 1l). Lags in the timing of frequency and duration peaks were evident—due to the fact that the duration of an event that crossed from one year to the next is assigned to the year in which the event started, e.g., a long-duration event in the eastern Tropical Pacific that lasted from 1997 into 1998, associated with the 1997/98 EL Niño, has its duration assigned to 1997.

The spatial influence of ENSO on the mean and variability of MHW properties was strongest in the eastern tropical Pacific (Supplementary Fig. 1). El Niño events tended to drive long-duration, high intensity MHW events there and with a higher frequency (Supplementary Fig. 1A, C, E). However, ENSO also increased the mean and variability of MHW duration in the northeast Pacific Ocean (Supplementary Fig. 1E, F), the variability of intensity off Western Australia and California (Supplementary Fig. 1D) and the variability of frequency over much of the Tropics in all ocean basins as well as the mid- and high-latitudes in the Pacific Ocean (Supplementary Fig. 1B).

There was also clear evidence of decadal and multi-decadal variability in the patterns of change in MHW properties. The satellite record (1982–2016) is too short to be able to differentiate multi-decadal climate variability from long-term trends. Over this period, the trend in the Pacific Decadal Oscillation (PDO) has been negative and the trend in the Atlantic Multidecadal Oscillation (AMO) has been positive (Supplementary Fig. 2C, D); the AMO and PDO patterns would thus be evident in maps of changes in SST, and therefore MHW properties, over this period as well. The changes in MHW properties (Fig. 1b, e, h, k) also clearly indicate signatures of a negative PDO pattern (SST decreases in the central and eastern tropical Pacific and in the eastern extra tropical Pacific Ocean; Supplementary Fig. 2A) and of a positive AMO pattern (SST increases in the North Atlantic particularly away from the mid-latitudes; Supplementary Fig. 2B). In fact, pattern correlations of these decadal change maps with the PDO/AMO patterns are −0.72/0.71 ($p < 0.05$) for SST and −0.59/0.61 ($p < 0.05$) for frequency; pattern correlations for intensity and duration were <0.40.

**SST trends drive marine heatwave changes**. The spatial patterns of changes in MHW frequency and duration were consistent with observed patterns of SST warming and cooling over the same period (Fig. 1k). At each location, we used a statistical climate model to simulate the expected trends in MHW properties (see 'Methods') given the observed secular SST trend (Fig. 3d). Results indicate that trends in MHW frequency could be explained by the trend in annual mean SST alone ($p < 0.05$) over more than 80% of the ocean surface area (Fig. 3a). The proportions were lower for

MHW intensity and duration, 59% and 53%, respectively (Fig. 3b, c). Areas where MHW trends were outside the range expected from the rise in mean SST alone (hatched areas in Fig. 3a–c), defined as excess trends, occurred in several contiguous blocks across the globe. Regions with excess trends might result from changes in SST variability. In fact, 61% of the global ocean with excess frequency trends corresponded to areas with significant trends in either the variance or skewness of SST (63% and 48% for intensity and duration, respectively; Fig. 3e, f). Areas exhibiting excess trends did not necessarily coincide with regions where severe MHWs have been documented. For example, at the locations of four of the recent high-impact MHWs described above, no excess trends were found off the coast of Western Australia or the northwest Atlantic while excess trends in MHW intensity were found for the northeast Pacific and the northern Mediterranean Sea (Supplementary Fig. 3).

**Long records from in situ stations**. Daily in situ measurements of ocean temperature at century-long monitoring sites were used to examine MHW properties over multi-decadal time scales. We selected six stations for which centennial-scale records (89–111 years) of daily ocean temperatures were available: Arendal (Norway), Port Erin (UK), Race Rocks (Canada) and Pacific Grove, Scripps Pier, and Newport Beach (USA; Table 1). Changes over time were calculated between early and late 30-year periods shared across all stations (1925–1954 and 1984–2013, respectively). All stations, except Newport Beach, exhibited annual mean SST warming of between 0.37 °C and 0.78 °C ($p < 0.05$); Newport Beach exhibited a non-significant change in SST (−0.003 °C, $p > 0.05$). There was clear centennial increase in annual MHW frequency at all stations (Fig. 4a–f, black lines). Changes between early and late 30-year periods shared across all stations (1925–1954 and 1984–2013, respectively) indicated an increase of between 0.40 and 1.8 annual events ($p < 0.05$ at all stations except Pacific Grove and Scripps Pier; Fig. 4a–f, red lines). Years without MHWs were also less common in the recent period: across the six stations, the proportion of years with no MHWs was 54–83% for the 1925–1954 period, reduced to 24–47% for the 1984–2013 period. This resulted in a shift from a predominance of years without any MHWs in the early period to a predominance of years with at least one MHW in the later period. No significant ($p > 0.05$) changes were identified for MHW intensity (Fig. 4g–l) or duration (Fig. 4m–r). It should be noted that, based on available data, these six sites were regionally biased to eastern boundaries of Northern Hemisphere oceans and specifically highlight the local nearshore changes in the north-eastern Pacific and northeastern Atlantic oceans.

**Extending the global record to the start of the twentieth century**. The global satellite-derived daily SST record we used is three and a half decades in length (1982–2016), and the observed patterns of change (Fig. 1b, e, h, k) contained signatures of internal climate variability. The century-long station records provide trends over a longer time scale but are regionally and coastally restricted. To extend our understanding of centennial MHW trends globally, we used multiple monthly averaged gridded (1/2° or coarser) in situ-based SST products to develop reliable proxies for MHW frequency and duration from the early twentieth century to the present. The data sets used were: HadISST v1.1[40], ERSST v5[41], COBE 2[42], CERA-20C[43] and SODA si.3[44], covering 1900–2016 (see 'Methods' for details). We developed a suite of proxies for observed MHW properties based on the relationship between daily derived MHW metrics and monthly averaged SST data using the six century-long in situ station-based temperature records, remotely sensed NOAA OI

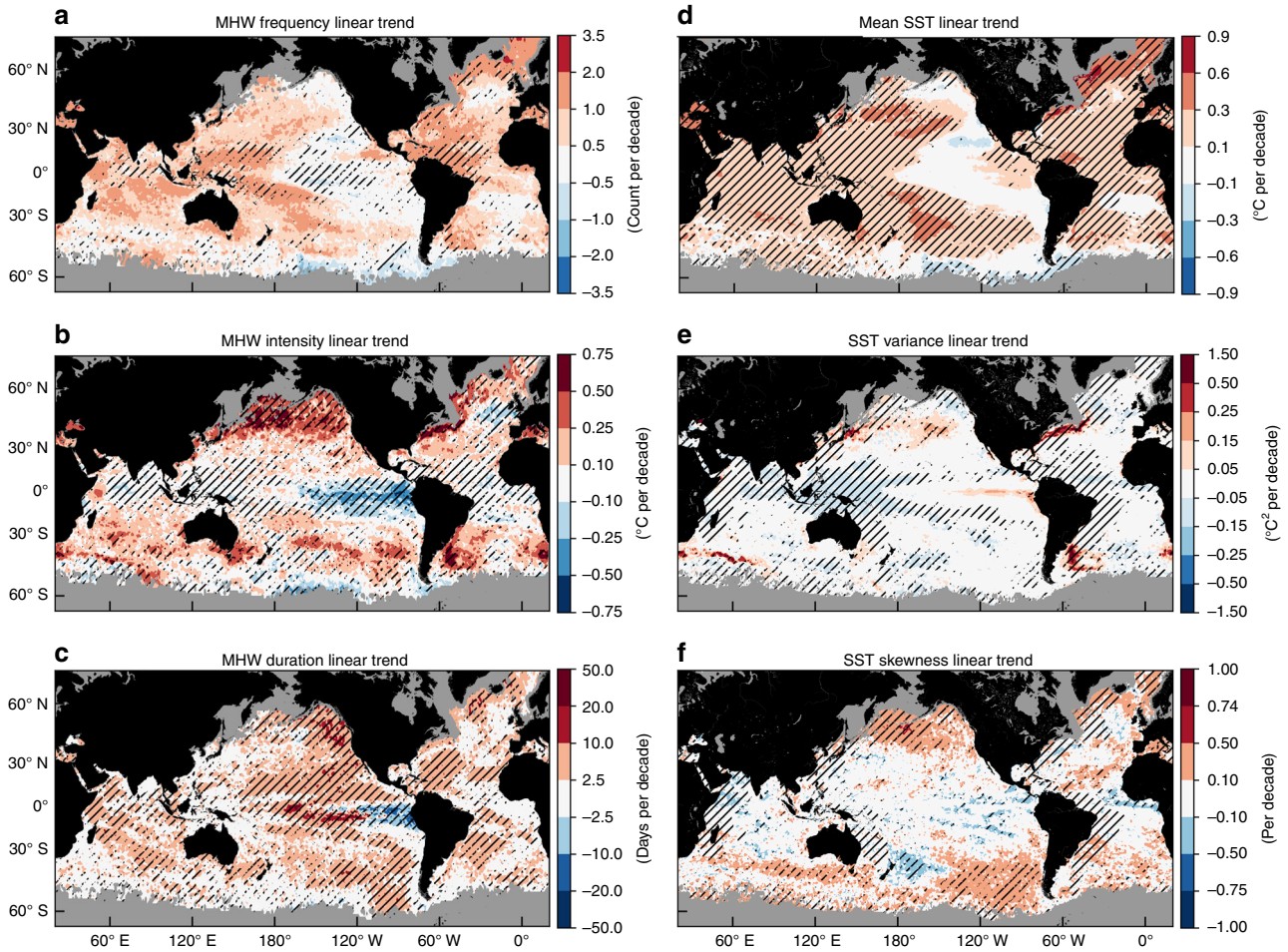

**Fig. 3** Identification of excess trends in marine heatwave properties globally. Shown are linear trends in **a** marine heatwave frequency, **b** marine heatwave intensity, **c** marine heatwave duration, **d** annual mean SST, **e** annual SST variance and **f** annual SST skewness from NOAA OI SST over 1982–2016. Hatching in **a**–**c** indicates the linear trend is significantly different from what is expected solely due to the rise in the annual mean SST (at the 5% level), based on the statistical climate model. Hatching in **d**–**f** indicates the linear trend is significant from zero ($p < 0.05$)

| Table 1 Century-long daily in situ records of ocean temperatures | | | | | |
| --- | --- | --- | --- | --- | --- |
| **Station** | **Time span** | **Number of years** | **Latitude** | **Longitude** | **% Complete** |
| Pacific Grove, USA | 1920–2014 | 95 | 36° 37.3′ N | 121° 54.2′ W | 94 |
| Scripps Pier, USA | 1917–2014 | 98 | 32° 52′ N | 117° 15.5′ W | 97 |
| Newport Beach, USA | 1925–2013 | 89 | 33° 36′ N | 117° 56′ W | 98 |
| Arendal, Norway | 1924–2016 | 93 | 58° 29′ N | 8° 47′ E | 95 |
| Port Erin, UK | 1904–2014 | 111 | 54° 5.1′ N | 4° 46.1′ W | 99 |
| Race Rocks, Canada | 1922–2016 | 95 | 48° 17.9′ N | 123° 32′ W | 98 |

The Norway data were obtained from the Havforskningsinstituttet Institute of Marine Research, the UK data from the Isle of Man Government Laboratory, the USA data from the Shore Stations Programme run by Scripps Institution of Oceanography and the Canada data from Fisheries and Oceans Canada. Note that partial initial and final years were rejected when determining the time span of each record

SST data, and the five gridded monthly SST data sets. We found that the annual count of months with SSTs above the 90th percentile could be used as a proxy for the annual average MHW frequency and the annual maximum monthly SST anomaly for the annual average duration of MHWs, respectively (see 'Methods' and Supplementary Note 1). This proxy-based approach allowed us to reconstruct annual time series of MHW frequency, duration and total MHW days globally from 1900 to 2016 (Fig. 5). MHW intensity was not extended in a similar way as no satisfactory proxy was found.

We examined changes in the proxy records for MHW frequency and duration between two 30-year periods: 1925–1954 (chosen to be consistent with the station analysis) and 1987–2016 (the last 30 years on record). Between 1925–1954 and 1987–2016, the proxy record showed increases in MHW frequency for 97% of the global ocean with most areas exhibiting changes of +0.3 to +1.5 annual events (Fig. 5a). An exception was the far North Atlantic Ocean which showed a decrease in MHW frequency, of a similar magnitude. As a global average, MHW frequency increased from about two events per year in 1900 to over three events in the last several years (Fig. 5b, black line). Between 1925–1954 and 1987–2016, there was a global average increase of 0.78 annual events ($p < 0.01$; a 34% increase). These results were also consistent with the significant increase in

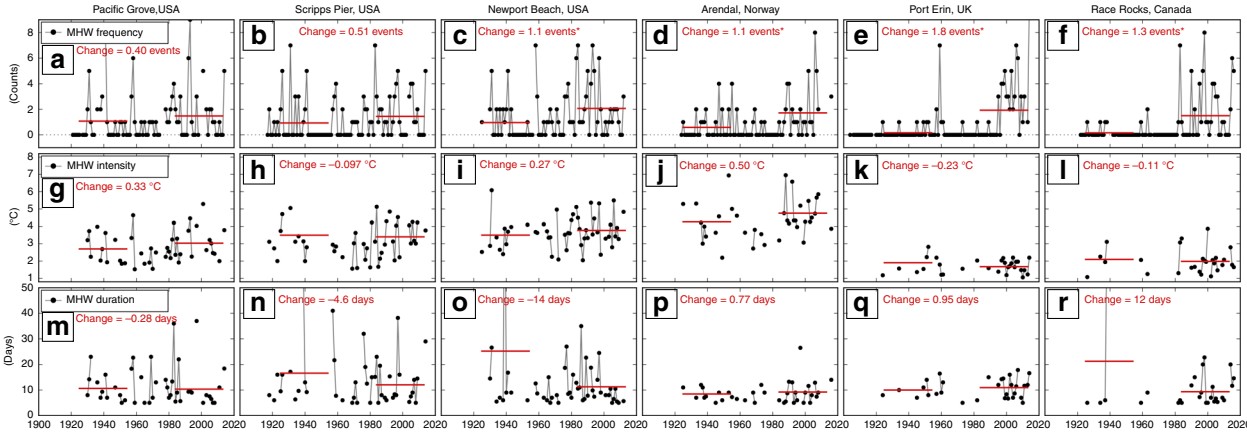

**Fig. 4** Marine heatwave properties at six sites with century-long daily in situ temperature records. Shown are **a**–**f** the number of marine heatwaves (MHW) in each year, **g**–**l** average annual MHW intensity, **m**–**r** average annual MHW duration at Pacific Grove, Scripps Pier, Newport Beach (USA), Arendal (Norway), Port Erin (UK) and Race Rocks (Canada). Red lines show means over 1925–1954 and 1984–2013 and red text indicates differences between these periods (a star indicates statistical significance at the 5% level)

**Fig. 5** Centennial-scale changes in marine heatwave proxies derived from monthly gridded sea surface temperature data sets. **a**, **c**, **e** Difference between 1925–1954 and 1987–2016 and **b**, **d**, **f** globally averaged time series of annual **a**, **b** marine heatwave (MHW) frequency, **c**, **d** MHW duration and **e**, **f** total MHW days based on monthly proxies derived from all proxy data sets (data set mean) over 1900–2016. In **a**, **c**,**e**, hatching indicates that all five data sets agreed on the sign of the change (corresponding to 5% significance based on a binomial distribution) and grey areas indicate no data. The equivalent maps for each individual data set can be found in Supplementary Fig. 4. In **b**, **d**, **f**, globally averaged time series of MHW properties are shown for (thick red line) the NOAA OI SST 1982–2016 data, (thin coloured lines) the individual proxy data sets and (thick black line) the mean across all five proxy data sets. The shaded areas show the 95% confidence intervals based on proxy model errors, averaged across the five data sets (taking into account temporal covariance)

MHW frequency observed in the few long (88–104 years) in situ daily station time series (Fig. 4).

Changes in the MHW duration proxy between the two periods showed an increase over 91% of the global ocean (Fig. 5c). The magnitude of the increase was typically up to 6 days but larger positive changes were found in the eastern tropical Pacific Ocean, northeastern Pacific Ocean and parts of the South Pacific Ocean (6–14 days). Globally averaged time series showed a general increase in MHW duration from about 11 days in the early twentieth century to over 15 days in recent years (Fig. 5d, black line). There was a globally averaged increase of 1.8 days between 1925–1954 and 1987–2016 ($p < 0.01$; a 17% increase). The increases in both MHW frequency and duration by the 1987–2016 period led to a globally averaged increase in the annual number of MHW days of 14 days ($p < 0.01$) from a baseline level of 26 days in 1925–1954 (Fig. 5f; a 54% increase).

The MHW proxy records exhibited signatures of variability on interannual to multi-decadal time scales. For example, there were peaks in globally averaged frequency and duration during El Niño years, consistent with the satellite record, as well as periods spanning several decades of raised or lowered values about the secular rising trend (Fig. 6, thick blue lines). As with the satellite record, we removed the influence of ENSO; in addition, the longer time series allowed us to remove the effects of two longer time scale modes, the PDO and the AMO (see 'Methods'). After the removal of this interannual, decadal and multi-decadal variability from the original time series, the long-term rising trend becomes more prominent (Fig. 6, black lines). Spatially, the effect of ENSO on MHW properties was primarily found in the tropical eastern and central Pacific Ocean. Despite such regionality, ENSO contributed globally to interannual variations in MHW frequency and duration on the order of 1–1.5 annual events and up to 15 days, respectively (Supplementary Fig. 5A–F). The PDO primarily affected the tropical and northern Pacific Ocean with variations in MHW frequency of up to 0.5 annual events and durations of up to 9 days (Supplementary Fig. 5G–L). The AMO primarily modulated MHW frequency in the tropical and far northern Atlantic Ocean by up to one annual events but also impacted MHW duration in parts of the Pacific by up to 6 days (Supplementary Fig. 5M–R). These patterns are consistent with the signatures of these modes in SST (Supplementary Fig. 2).

The impact of modes of variability alone can be isolated, assuming linearity in our approach, by comparing the original time series and the time series with the modes removed (e.g., difference between blue and black lines in Fig. 6). The combination of these three modes explains 18%, 56% and 36% of the variance of the globally averaged time series of MHW frequency, duration and total days, respectively. After the removal of these modes, the changes in globally averaged MHW frequency, duration and total days between 1925–1954 and 1987–2016 were similar to the changes derived without the removal of these modes: +0.68 events, +1.8 days and +12 days, respectively. Therefore, while these climate modes clearly have a large impact on the variability of globally averaged MHW metrics, and this transient variability certainly leads to MHW events and ecological impacts, their removal highlights the secular trend and further supports our conclusion regarding significant MHW frequency and duration increases on centennial time scales.

Extraordinary global ocean warming occurred during the 2014–2016 period. Contributions to this warming included the 2014–2016 marine heatwave in the NE Pacific, the 2015/16 El Niño, the 2014 switch to a positive PDO phase and overall background warming[45]; this prolonged warming was particularly evident as it came after a period of global warming hiatus. This 2014–2016 strong warm anomaly was also evident in our globally

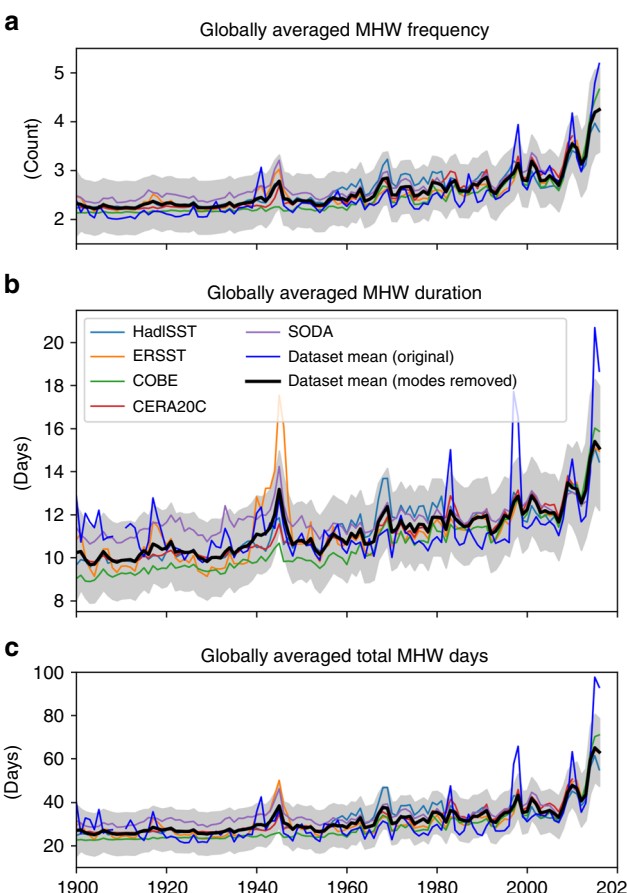

**Fig. 6** Influence of climate modes on marine heatwaves globally. Globally averaged time series of annual (first row) marine heatwave (MHW) frequency, (second row) duration and (third row) total MHW days based on monthly proxies over 1900–2016. Shown are the original results which include the influence of major global-scale climate modes (i.e., ENSO, PDO, AMO; blue lines) and results after the removal of their global signature (thin coloured lines and thick black lines). The shaded areas show the 95% confidence intervals based on model errors, averaged across proxy data sets (taking into account temporal covariance; after removal of climate modes). Note that the blue lines in this figure are identical to the black lines in Fig. 5

averaged time series of MHW statistics (Fig. 6, blue lines). While the removal of ENSO and the PDO reduced the magnitude of this warming (Fig. 6, black lines), consistent with the roles of these modes noted above, the warming signal was still evident. This was primarily due to the persistent marine heatwave in the northeast Pacific[6], which was unprecedented in the historical record and cannot be fully explained by the coincident occurrence of climate modes at the time (e.g., ref.[46] for the related high-latitude Pacific warming of 2016).

## Discussion

We have characterised MHW trends and variability globally from 1900 to 2016 using a multi-data set approach and a unified MHW framework. We have shown that between 1925–1954 and 1987–2016, on global average, MHW frequency increased by 34% and the average MHW duration increased by 17%. Together, the combined changes in MHW frequency and duration amount to a 54% increase in the total number of annual MHW days. Much of this change has occurred over the last several decades indicating that the warming trend accelerated over the 1925–2016 period. Changes in mean SST can explain trends in MHW properties

over most of the global ocean, particularly for frequency. Shifts in internal temperature variability, measured through SST variance and skewness, are also occurring and contribute to much of the MHW trends observed over the remainder of the global ocean, particularly for MHW duration and intensity. Modes of climate variability contribute to significant MHW variations both regionally, and globally, but do not greatly affect the centennial-scale secular changes described above.

These significant increases in MHW properties between 1925–1954 and 1987–2016 have coincided with shifts in species' distributions and changes in biodiversity patterns[6,8,10,12,13,15]. It is evident from regional-scale studies that MHWs can cause widespread loss of habitat-forming species such as kelps and corals, drive shifts in species distributions, alter the structure of communities and ecosystems, and have economic impacts on aquaculture and seafood industries through declines in important fishery species. MHW impacts on marine species and ecosystems can occur on a range of time scales, with some species showing effects after a few days and others responding only after several months of elevated temperatures—and these impacts can last beyond the duration of the event itself[10]—responses that are also confounded by the thermal tolerance of different species living in the same region[47]. Such ecological impacts are likely to have become more prevalent with the increasing frequency and duration of MHWs over the last century.

Given the likelihood of continued ocean surface warming throughout the twenty-first century[20], we expect a continued global increase in MHW frequency and duration in the future with implications for marine biodiversity and the goods and services ocean ecosystems provide. Like the effects of atmospheric heatwaves on terrestrial ecosystems, which include widespread losses of crops and forests and reductions in biodiversity[48], the impacts of MHWs have had major impacts on societal interactions with the ocean—specifically in terms of fisheries, aquaculture and tourism[11,14,15]. Improving the spatial coverage, temporal resolution and survey design of monitoring programmes, supporting citizen science observation campaigns (e.g., www.redmap.org.au) and maintaining long-term records, both in situ and remotely sensed, will help facilitate improved detection and attribution of the ecological impacts of MHWs[49,50] and is the only way continued changes can be quantified[51]. In addition, documenting events with a consistent framework[1] across space and time will enable the comparison of the physical properties of different MHWs and contribute towards a greater understanding of their distribution and drivers. Given the pronounced warming trend coinciding with notable MHWs in recent years, we highlight the increasing need for rapid reporting of these events as they emerge and the ecological consequences witnessed by scientists and the broader community.

## Methods

**Defining marine heatwaves.** We identified MHWs from daily SST time series and calculated metrics to characterise their frequency, intensity and duration. A MHW was defined following ref. [1] as a discrete prolonged anomalously warm water event. 'Discrete' was defined quantitatively as an identifiable event with recognisable start and end dates, 'prolonged' meant a duration of at least 5 days and 'anomalously warm' was defined by reference to a baseline, seasonally varying threshold. Heatwave events were found by identifying periods when daily temperatures were above the seasonally varying 90th percentile (threshold) for at least five consecutive days. Two events with a break of less than 3 days were considered as a single event. The 90th percentile was calculated for each calendar day using daily SSTs within an 11-day window centred on the date across all years within the climatology period, and smoothed by applying a 31-day moving average. The choices of an 11-day window and a 31-day moving average were motivated to ensure sufficient sample size for percentile estimation and a smooth climatology[1]. A seasonally varying threshold allows identification of anomalously warm events at any time of the year, rather than events only during the warmest months.

Each MHW event had a set of properties including duration and several measures of intensity (see ref. [1] for more discussion). We considered metrics for duration (time between start and end dates) and maximum intensity (maximum temperature anomaly, relative to the seasonally varying climatological mean, over the event duration). We then calculated time series of the annual average duration and annual maximum intensity across events in each year, and also frequency (defined as the number of events occurring in each year) and the total number of MHW days in a year. Note that when calculating the annual statistics of events which occur across several years, the duration and intensity are assigned to the start year of that event. The statistical significance of the long-term trends presented in the main text (described below) is not sensitive to the choice of percentile threshold (90th, 95th and 98th percentiles tested; not shown), and therefore, our conclusions about changes in MHW properties are robust to the choice of threshold.

This definition builds on recent developments in defining atmospheric heatwaves[52] and was developed further and adapted for the marine environment by a cross-disciplinary team consisting of atmospheric scientists, oceanographers and marine ecologists[1]. The MHW definition as used in this manuscript is available as software modules in Python (http://github.com/ecjoliver/marineHeatWaves) and R (https://github.com/ajsmit/RmarineHeatWaves).

**Daily, global, remotely sensed SSTs covering 1982–2016.** We used the National Oceanic and Atmospheric Administration (NOAA) Optimum Interpolation (OI) Sea Surface Temperature (SST) V2.0 high-resolution gridded data set[37,38] to derive and calculate MHW properties globally. This data set was derived from remotely sensed SST by the advanced very high resolution radiometer (AVHRR), and it should be noted that retrievals from high-cloud covered areas may need to be treated with caution[53,54]. The data have been interpolated daily onto a $0.25° \times 0.25°$ spatial grid with global coverage from 1982 to 2016. We calculated MHW properties and annual time series for the period 1982–2016 from the raw daily SSTs at each grid point globally, and used a 30-year subset (1983–2012) to define the baseline climatology and 90th percentile threshold. For each of the annual MHW time series, we calculated (i) a mean value at each grid point, (ii) the difference between the mean values over the first and second halves of the data (1982–1998, 2000–2016) and (iii) a globally averaged (area-weighted) annual time series, including linear trend. Note that the point where we split the data (1999) was chosen to have two equal-sized samples (17 years); the results are not strongly sensitive to this choice. We excluded any grid cells from the analysis which had continuous ice cover duration for longer than 5 days—gaps of 5 days or less were interpolated using the MHW algorithm.

We did not calculate linear trends at each location individually since the time series of MHW metrics have certain properties for which standard trend-fitting techniques may have unusual biases, e.g., they are bounded at a lower value (zero for frequency, 5 for duration) and some are quantised. We instead present differences of the mean value between two time slices, making no assumptions about the distribution of the data. Statistical significance of the time slice differences was determined using a two-sample Kolmogorov–Smirnoff (K-S) test, a non-parametric test allowing for non-normally distributed data. Concerns around the use of linear trends are less relevant for the globally averaged time series; however, ordinary least squares estimates of linear trends may be biased due to the presence of outliers or non-normally distributed data. We therefore calculated linear trends of the globally averaged time series using Theil–Sen (TS) estimates[55]. A TS estimate of the linear trend is more robust for time series data that are heteroskedastic or have a skewed distribution. Statistical significance of trends was estimated using a 95% confidence interval.

Given the relatively low threshold used, the MHW definition will encompass events from those with low intensity that are likely to have limited impact, through to the most extreme events that would likely have major impacts. Under this criteria, there were typically 1–3 events annually at most locations (Fig. 1a). This definition is consistent with commonly used atmospheric heatwave definitions, and provides sufficient samples of MHW events to provide robust statistics using standard tests.

We removed the influence of ENSO from the SST time series, before the MHW detection, using a statistical approach. We first estimated the ENSO signal at each pixel by regressing daily SSTs onto the multivariate ENSO index (MEI[56]) and subtracted the linear prediction based on this model. We included monthly leads and lags of the MEI up to ±1 year, into a multiple linear regression model. The MEI is defined monthly as a 2-month average (Dec–Jan, Jan–Feb, etc.) and we assumed the monthly values to be centred on the middle of second month. While this will implicitly impose a 15-day lag on the MEI with respect to the true central date, which should be the end of the first month/beginning of the second month, we are not greatly concerned with how the regression is distributed across lags and do not expect this 15-day difference to impact our ±1 year lead-lag analysis. We then identified the MHWs from the ENSO-less SST time series as above, except we used the climatological mean and threshold based on the original SST (i.e., that which includes ENSO). The original climatology was used because what we consider MHWs, and what ecosystems are adapted to, are based on the real-world threshold and we are interested in how ENSO changes the frequency, intensity and duration of exceedances over this threshold. Note that this method has two assumptions: (1) a linear relationship between the MEI and SST and (2) that the MEI captures all of the ENSO signal. The MEI was chosen as it takes into account the full-scale atmospheric and oceanic response to ENSO. Both assumptions lead to an imperfect model and therefore a small residual ENSO signal may remain in the data.

**Daily in situ sea temperature from six monitoring stations**. We obtained daily in situ measurements of ocean temperature at six century-scale monitoring stations (see Table 1). We were unaware of any other stations with daily measurements over a period of more than 80 years, with few missing values or large data gaps. If individual daily records were missing, we interpolated over gaps of up to 5 days (the minimum threshold for MHW duration[1]) and otherwise flagged the entire year of data as missing. The data for Scripps Pier were bias-corrected by −0.45 °C post-1988 as recommended in ref. [57]. MHW properties and annual time series and linear trends were calculated as for the NOAA OI SST data with the same 1983–2013 period used to define the baseline climatology and threshold. We calculated differences of mean MHW properties between the earliest and latest 30-year periods shared across all stations: 1925–1954 and 1984–2013.

**Proxies for MHW properties based on monthly SSTs**. In the absence of centennial-scale global gridded daily SST data, monthly averaged gridded SSTs can be used to evaluate changes in certain MHW characteristics globally in a longer temporal context. Monthly SSTs cannot resolve many individual MHWs directly, since many events are shorter in duration than 1 month. However, we have used monthly SSTs to develop proxies for annually aggregated MHW properties. This was done using both the long station time series and global monthly SST data sets. We developed proxies by selecting the MHW metrics we wished to predict (annual frequency, duration, intensity) and the set of variables which may be possible predictors (annual mean SST, annual maximum SST, annual count of months above a threshold, etc). Generalised linear models were trained on the long station series over the post-1982 period, and validated over the pre-1982 period. This was used to select which variables should be used as predictors for the annual MHW metrics. Then, these variables were used to train proxy models on the post-1982 MHW satellite data using predictors derived from the monthly SST analyses. Monthly SST analyses have no daily data with which to validate the model selection —which is why the model selection was performed based on the long station time series, and why the use of the stations is important to the study.

**Proxies based on daily station time series**. Monthly averages of the daily temperatures at the six stations were calculated and from these data we calculated a monthly climatological mean and 90th percentile across all years. We then calculated eight annual quantities: TMM was the annual mean SST, TMX was the annual maximum monthly SST, TAM and TAX were the annual mean and maximum monthly SST anomaly (relative to the monthly climatological mean), TTM and TTX were the annual mean and maximum anomaly (relative to the 90th percentile threshold) over all months which exceed the 90th percentile, and NA and NT were the annual count of months above the seasonally varying climatological mean and 90th percentile, respectively. Note TAM and TAX (TTM and TTX) had a missing value if NA (NT) was zero in that year.

Annual MHW properties, derived from daily temperature time series (as described above), were modelled using a generalised linear model (GLM)[58] with monthly based annual proxies as predictors:

$$g(y) = \beta_0 + \beta_1 x + \varepsilon \qquad (1)$$

where $y$ is an annual MHW property (frequency, intensity, duration), $x$ is one of the proxies listed above, $\beta_0$ and $\beta_1$ are the regression coefficients and $g$ is the link function. GLMs are not restricted to normally distributed data and $y$ is assumed to be distributed according to the exponential family (which includes the normal and Poisson distributions). Models were trained over the available data at each station since 1982 and fit using iteratively reweighted least squares. The fitted models were used to predict MHW properties over the entire time series and validated using the independent data prior to 1982. We tested each of the eight proxies as predictors for each of the three MHW properties (see Supplementary Note 1).

Distributions and link functions were chosen as follows. MHW frequency and duration are count data and were modelled by a Poisson distribution, intensity was approximately normally distributed and was modelled by that distribution accordingly. The proxies NA and NT could also be considered count data and an identity link function was used when predicting frequency or duration, a log link function otherwise. For the remaining proxies, which were approximately normally distributed, a log link function was chosen when predicting frequency or duration, an identity link function otherwise. Proxies were found for frequency and duration, but intensity was too poorly predicted to provide any proxy (see Supplementary Note 1, Supplementary Table 1 and Supplementary Fig. 6).

**Proxies from the global data sets of monthly SSTs**. To examine MHWs globally over centennial time scales, we used the Hadley Centre Sea Ice and SST v1.1[40] (HadISST), NOAA Extended Reconstructed SST v5[41] (ERSST), Centennial in situ Observation-Based Estimates 2 SST[42] (COBE), Coupled European Centre for Medium-Range Weather Forecasts (ECMWF) ReAnalysis 20C SST[43] (CERA-20C) and Simple Ocean Data Assimilation si.3 SST[44] (SODA) data sets to derive MHW proxies from 1900 to 2016. Note that HadISST, ERSST and COBE end in 2016 while CERA-20C ends in 2010 and SODA ends in 2013. The HadISST, ERSST and COBE data sets consist of monthly gridded data in which in situ observations (and remotely sensed satellite observations, after 1982) have been interpolated onto a regular horizontal grid (1° × 1° for HadISST and ERSST; 2° × 2° for COBE). The

CERA-20C data set is a global coupled reanalysis which relaxes the air-sea interface towards HadISST v2 SSTs and data have been obtained on a 1° × 1° horizontal grid (note that version 2 of HadISST was not directly available at the time of writing, which is why we used version 1 above). The SODA data set is a global ocean reanalysis that assimilates individual observations from the International Comprehensive Ocean-Atmosphere Data Set (ICOADS) Release 2.5 and provides data on a 1/2° × 1/2° horizontal grid. We calculated the eight proxies defined in Section 4.1 at each location and fitted the models by Eq. 1 to the NOAA OI SST annual MHW properties, area-averaged over the above-listed data sets' grid cells, from 1982 to 2016. Results demonstrated that monthly average temperatures could be used as proxies for MHW count (through NT) and duration (through TAX) but not for intensity (see Supplementary Note 1, Supplementary Figs. 7–8).

We used the full record of global, gridded, monthly SST data to reconstruct MHW frequency and duration and to analyse centennial-scale trends in those properties. We calculated annual mean MHW frequency and duration from 1900 to 2016. We used the product of MHW frequency and duration to calculate the total number of MHW days per year. Due to the sparse coverage and large uncertainties in early years[35], we restricted the analysis period to data since 1900; for the remaining data, any year in which at least one month of data was missing was marked as missing. We also omitted from the data sets any grid cell which repeated the seasonal cycle at least 10 times which is common in the Southern Ocean due to low data coverage. Correlations between the annual MHW metrics from the long station time series and the proxies indicate good agreement regionally over the overlapping time periods (Supplementary Fig. 9). The correlations shown in Supplementary Fig. 9 have been averaged across the five proxy data sets using the Fisher's z-transformation technique of ref. [59].

We calculated spatial maps of the mean and the difference between the 1925–1954 period (consistent with the early period chosen for the long station data) and the 1987–2016 period (the last 30 years in the time series), as well as a globally averaged annual time series and its time slice difference. Prior to this the five global proxy data sets were regridded onto a shared grid (2° × 2°) and then an average across all data sets was performed (data set mean). Globally averaged time series from the proxies were adjusted to have the same mean value as the NOAA OI SST time series over the 1982–2016 period. Note that for years in which data were missing over more than half of the area to be averaged, global average time series values were not calculated and were marked as missing data. Globally averaged time series and trends were not strongly sensitive to the exclusion of ENSO (see Method below), unlike the 35-year satellite-derived daily SST data. This suggested that longer time series reduced the influence of interannual climate variations by exposing more of the longer-term trend.

SST in parts of the ocean, e.g., the Southern Ocean, the South Indian Ocean, and much of the Pacific Ocean, were sparsely observed prior to the mid-twentieth century. The global monthly data sets are statistically or dynamically interpolated and so provide data for all available space and time ocean grid cells. However, the Hadley Centre SST data set[60,61] (HadSST3, v3.1.1.0) is not global in coverage: rather than interpolating over all space and time coordinates it consists of spatial means within 5° × 5° bins, leading to missing values in the absence of data. We tested the impact of spatial bias by examining global trends in MHW proxies derived from HadSST3 data. We rejected any years in which there were missing months from the HadSST3 data analysis and we also calculated the global averages only for years in which at least half of the ocean surface consisted of valid data. Despite these restrictions: the global data sets (presented in Fig. 5), which may be subject to artefacts from having values in regions with no observations, and HadSST3 provided consistent global trends in MHW properties (Supplementary Fig. 10). We have considered this, along with the inclusion of the long in situ station data and the NOAA OI SST data since 1982, as a multi-data set approach.

We removed the influence of ENSO, the Pacific Decadal Oscillation (PDO)[62] and the Atlantic Multidecadal Oscillation (AMO)[63] from the MHW proxies using a statistical approach. First, we fitted the proxy model to the original SST data. Then, we generated proxies with this fitted model, using as predictors the SST time series where the components linearly related to the MEI, PDO and AMO were removed (by linear regression, as above). We allowed for monthly leads and lags of the MEI up to ±1 year; the PDO and AMO indices were smoothed with a 5-year running average and only the zero-lag relationships with SST were considered. This method assumes a linear relationship between the climate modes and SST, and also that these indices capture all of the ENSO, PDO and AMO signals. Both assumptions lead to an imperfect model and therefore residual climate variability may remain in the data. We used the NOAA Earth System Research Laboratory monthly unsmoothed AMO index and the NOAA National Centers for Environmental Information monthly PDO index. Note that the MEI and the PDO index were not independent ($r = 0.32$) and so the component of the PDO index linearly related to the MEI was first removed by linear regression; neither the MEI or the PDO index are significantly correlated with the AMO index, and so there is no need to perform a similar correction to the AMO index.

**Marine heatwave proxy uncertainty**. We have quantified the uncertainty in the globally averaged proxies by considering two sources: the proxy model and the observations themselves. Model prediction uncertainty was quantified by assuming the model parameters were distributed normally, with a variance given by the uncertainty on the model fit, and performing a Monte Carlo resampling to produce

many new predictions from which a 95% confidence interval was calculated. A Monte Carlo sample size of 200 was used for the long time series data, separately for each of the five proxy data sets. This provided a model error variance $\sigma_{\text{mod},i}^2$ at each grid cell $i$. The error on the global mean $\sigma_{\text{mod}}^2$ was calculated by propagating the point location errors using the following formula, which takes into account the spatial covariance of MHW properties,

$$\sigma_{\text{mod}}^2 = \left[\sum_i \sigma_{\text{mod},i}^2 + 2 \sum_i \sum_{j>i} \text{cov}(y_i, y_j)\right] / n^2 \quad (2)$$

where $y_i$ is the MHW proxy at grid cell $i$, cov indicates the covariance between two variables and $n$ is the number of valid grid cells. The model error on the data set-mean time series (Fig. 5b, d, f, shaded areas) was calculated by propagating the $\sigma_{\text{mod}}^2$ for the five data sets and including the covariance of the time series using a formula equivalent to Eq. 2.

Observational uncertainty was unavailable from any of the five global data sets considered. However, the HadSST3 data set does include a quantification of observational errors. At each location the measurement and sampling uncertainty provided with the HadSST3 data was combined with a Monte Carlo technique to simulate many alternate proxy time series (sample size of 200 at each grid cell). This resulting error on the proxies was defined as the observational error $\sigma_{\text{obs},i}^2$. This enabled us to estimate the influence of nonhomogeneity in the variance which might arise due to data errors and sampling density. The uncertainty in the global mean due to observational uncertainty $\sigma_{\text{obs}}^2$ was calculated as above, and the total error due to both sources $\sigma_{\text{tot}}^2$ was then calculated as follows:

$$\sigma_{\text{tot}}^2 = \left[\sum_i \sigma_{\text{mod},i}^2 + \sum_i \sigma_{\text{obs},i}^2 + 2 \sum_i \sum_{j>i} \text{cov}(y_i, y_j)\right] / n^2 \quad (3)$$

which assumes the two sources are independent. Model, observational and total errors for the HadSST3 proxies are shown in Supplementary Fig. 10.

**Calculating excess trends in MHW properties**. Trends in MHW properties arise due to a trend in the mean SST, a trend in higher-order SST statistics (i.e., variability), or both. The variability of SST at a given location, region or basin can be forced by local to large-scale climate variability, e.g., persistent atmospheric patterns or shifts in large-scale climate modes over century-long time scales under anthropogenic climate change. Changes to variability on centennial time scales have been demonstrated in the atmosphere and are distributed non-uniformly over the globe[64–66]. Changes to ocean variability have been observed, primarily through the mesoscale eddy field as observed through satellite altimetry[67,68] and these changes will influence the extremes. For example, off southeastern Australia projections indicate an increase in eddy variability with anthropogenic climate change[69] and an associated increase in SST extremes from increases in both the mean and variability of SST[70].

We developed a statistical model to simulate trends in MHW properties due solely to a trend in the mean SST. To do so, we simulated a SST time series which assumes its statistical properties (mean, variance, autocorrelation) are stationary in time. This was accomplished using a stochastic climate model based on the concept that ocean temperature variability is a slow dynamical system, a red noise signal, generated by integrating stochastic atmospheric forcing, or white noise[71]. A model of SST variability was implemented in which the ocean is treated as a motionless mixed layer forced and damped by stochastic surface heat fluxes[72]. This model has been used successfully to capture SST variability in the North Pacific[73,74] and the North Atlantic[74,75]. This is expressed as a first-order, autoregressive process:

$$T_{t+1} = aT_t + \varepsilon_t \quad (4)$$

where $T_t$ is daily SST at time (day) $t$, $a$ is the autoregressive parameter and $\varepsilon_t$ is a white noise process with zero mean and variance $\sigma_\varepsilon^2$ (ref. [76]). The autoregressive parameter can be expressed as a time scale $\tau = -1/\ln(a)$, in days. The simulated temperature series has a mean of zero with no secular trend. Given a time series of observed SST, after removing the seasonal climatology and linear trend, the model parameters were determined by ordinary least squares regression on the lagged SST with itself. The seasonal climatology was removed to be consistent with the MHW definition; and the linear trend was removed so that we could later prescribe one. The model fit to the daily NOAA OI SST data (1982–2016) can be found in Supplementary Note 2 and Supplementary Fig. 11. Higher-order autoregressive models could be considered but the model selection procedure required was beyond the scope of the present study.

We then generated simulated SST time series at each location by using the spatially varying fitted parameters, random white noise data $\varepsilon_t$ covering 35 years (i.e., 1982–2016), and specifying a constant linear trend. We then applied the MHW definition to these simulated $T_t$ time series and calculated the trends in annual MHW properties. This was undertaken for $N_\varepsilon$ independent realisations of $\varepsilon_t$ and from this we calculated a 95% confidence interval on the trends in MHW properties. Note that the statistical properties of the SST time series pertaining to short-term variability remained stable over the entire record, only the mean SST was allowed to vary. Therefore, this confidence interval provided the range of trends that we expect solely from a change in the mean SST itself. We then

compared the actual trends in annual MHW properties and if they lay outside this confidence interval we indentified these trends as excess trends ($p < 0.05$). In order to facilitate this excess trend calculation, we relaxed the restriction on using linear trends at individual grid cells, which was less of a concern over shorter time scales (approx. 30 years), and use the Theil–Sen estimator instead of ordinary least squares to avoid problems of non-normality in the data.

**Code availability**. The code used to analyse these data and generate the results presented in this study can be obtained from https://github.com/ecjoliver/Global_MHW_Trends (doi: 10.5281/zenodo.1188863). All figures were created using the software package Python, specifically the the matplotlib and basemap modules (https://matplotlib.org/, https://matplotlib.org/basemap/). The coastline data are the Global Self-consistent, Hierarchical, High-resolution Geography (GSHHG) Database (https://www.soest.hawaii.edu/pwessel/gshhg/), which has been released under the GNU Lesser General Public License, and is provided with the basemap Python module.

**Data availability**. We have used publicly available data only; new data were not generated as a result of this study. NOAA high resolution SST and COBE-SST2 data were provided by NOAA/OAR/ESRL PSD (Boulder, CO, USA) from their website (www.esrl.noaa.gov/psd/). Hadley Centre SST data were provided by the Met Office (UK) from their website (www.metoffice.gov.uk/hadobs). The in situ Norway data were obtained from Jon Albretsen of the Flødevigen Research Station, Havforskningsinstituttet Institute of Marine Research, the UK data from the Isle of Man Government Laboratory, the USA data from the Shore Stations Programme run by Scripps Institution of Oceanography and the Canada data from Fisheries and Oceans, Government of Canada.

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

## Acknowledgements

This contribution is an outcome from the working group 'Marine Heatwaves – physical drivers and properties' hosted at the University of Western Australia (UWA) Oceans Institute and led by T.W., D.A.S., N.J.H. and E.C.J.O (www.marineheatwaves.org). The workshop received funding support from a UWA Research Collaboration Award, a UWA School of Plant Biology synthesis grant and the Australian Research Council (ARC) Centre of Excellence for Climate System Science (ARCCSS). The workshop, and this paper, makes a contribution to the interests and activities of the International Commission on Climate of IAMAS/IUGG and the World Climate Research Programme Grand Challenge on Extremes. This is a PMEL contribution number 4415. S.E.P.-K. was supported by ARC grant DE140100952, M.G.D. by ARC grant DE150100456, L.V.A. by ARC grant CE110001028, D.A.S. by NERC IRF NE/K008439/1 and T.W. by ARC grants FT110100174 and DP170100023. M.T.B. was supported by NERC grant NE/J024082/1, J.A.B. acknowledges support from the ARCCSS (CE110001028), E.C.J.O. by ARC grants FS110200029 and CE110001028, N.J.H. acknowledges funding support from the ARC Centre of Excellence for Climate Extremes (CE170100023) and the National Environmental Science Programme (NESP) Earth Systems and Climate Change (ESCC) Hub Project 2.3 (grant B0024391), P.J.M. by Marie Curie CIG PCIG10-GA-2011-303685 and NERC grant NE/J024082/1, S.C.S. by an Australian Government RTP Scholarshipin, M.F. by a CAS-CSIRO collaborative project on Marine Science and the Blue Economy and the Western Australia Marine Science Institution.

## Author contributions

E.C.J.O. led and coordinated the various components of the study throughout. E.C.J.O., M.G.D., M.T.B., P.J.M. and T.W. led the initial design of the study. All authors (E.C.J.O., M.G.D., M.T.B., P.J.M., T.W., D.A.S., L.V.A., J.A.B., M.F.., A.S.G., A.J.H., N.J.H., S.E.P.-K., H.A.S. and S.C.S.) discussed the results, aided in their interpretation and contributed in writing the paper.

## Additional information

**Competing interests:** The authors declare no competing interests.

