## [Peer Review File(PDF 771 kb) · Nature Communications]

Reviewers' comments:

Reviewer #2 (Remarks to the Author):

This study investigates changes in marine heat wave characteristics over the last century based on a multi-dataset approach using satellite and in-situ measurements of SST. The authors apply the marine heat wave framework developed by Hobday et al. and find a significant increase in global heat wave characteristics, modulated by strong interannual-to-decadal variability.

The authors address a very important and timely topic. The manuscript is very well written and I find no major flaws in the results and methods. Congratulations. However, I have a few comments that prevent me to accept the manuscript in this present form. I am happy to accept the manuscript if the authors can adequately address my concerns raised below.

Major points:

- I think the title as it currently stands is too general and in fact it has not been demonstrated properly (using a proper attribution framework such as described by Stott et al. 2004) that (anthropogenic-induced) ocean warming is responsible for more frequent and longer marine heat waves. In addition, the title should indicate that the authors investigate changes over the satellite period and the past century and not future changes. This is not clear with the current title.
- What is the sensitivity of the results to the definition of heatwaves. The authors currently just apply one possible definition of heat waves (i.e. 90th percentile and at least five consecutive days). How would the results change when a higher percentile would be considered? I think this could be easily tested, quantified and discussed.
- Also, what is the motivation behind the criteria 'for at least five consecutive days'? Is there any literature available that justifies this criteria for all marine species? In any case, what is the average duration of a heat wave when just the 90th percentile is applied? Isn't it anyway longer than 5 days?

Minor points:

L.84-92: This paragraph seems to summarize all results and reads like an abstract. It does not really fit here.

I. 366-367: It is not clear to me how the climatological baseline from the 1983-2012 period was calculated. Did you detrend the data before calculating the 90th percentile?

I.388-389. What do you mean with 'moderately warming anomaly events'? Please clarify.

I.390-391: Yes, but if they are not rare events, they may not have an impact on organisms and ecosystems? In light of this, can you still justify your text in the introduction that states that especially extreme events may have significant impacts on marine ecosystems?

I. 326-328: I somehow disagree with this statement. I think a proper risk assessment for all marine species cannot be done with only one definition for marine heatwaves. The vulnerability of species to temperature changes is likely species dependent. Therefore, a unified framework will not help that much for a risk assessment.

Reviewer #3 (Remarks to the Author):

Many thanks for providing me with the opportunity to review the submission by Oliver and co-authors. I thoroughly enjoyed reading this new contribution, which discusses the topic of heatwaves in the marine environment. I must apologise to the Editor and the Authors for my tardy review.

The presented analysis provides a good overview for the reader, however I believe a number of key points and opportunities to link the present work to existing heat wave literature have been missed in the current version. I also think the text should more comprehensively cover existing literature, in which it appears considerable work has been already published. I also found some of the text wordy, and difficult to parse. The manuscript would benefit from a start to finish re-read to improve flow.

I have raised a number of queries about the selection of datasets, and raise the question of whether dataset selection matters to the key conclusions of the present work. While the MetOffice (HadISST and HadSST) datasets do benefit from considerable quality control, due to data sparsity subjective decisions are made during dataset construction. The impact of these decisions is not always obvious, but can be evaluated when multiple observational products are compared and contrasted alongside one another.

In addition to the general comments above, I have added a number of specific points below for consideration by the authors.

I believe that the present submission is relevant to a wide readership, and with further refinement should be published in the Nature Communications. Considering my comments above, and specific comments below, I suggest major revisions and look forward to reading an improved manuscript in press in the coming months.

Specific Comments

Page 3, lines 34-44: I would recommend a tightening up of the abstract, the contents of this paper are really an interesting read, yet the abstract didn't "sell" it to me

Page 4, lines 51-52: Don't bury the lede. Reverse this sentence, this happened, because, so.. "A redistribution of marine species, and reconfiguration of ecosystems has occurred due to pervasive increases to global SSTs..". There are a number of instances of reversed sentences throughout

Page 4, lines 57-58: To point out to a reader that these extreme events AND their impacts really matter, I suggest augmenting to introduce some of the atmospheric heat wave literature (e.g. Meehl & Tebaldi, 2004; Poumadere et al., 2004; McMichael et al., 2006; Barriopedro et al., 2011; Mitchell et al., 2016 for example and many others) and the very large mortality impact of these events. Extremes events do matter! Now we're looking at the ocean...

Page 5, lines 79-83: Which versions of datasets were used?

For OISSTv2, there is an AVHRR and an AVHRR+AMSR-E version of the data (extending from 2002-11), the Banzon et al (2016) citation is more up-to-date than Reynolds et al (2007).

For MetOffice products, the latest HadISST2 dataset is v2.2.0.0, and HadSST3 is v3.1.1.0.

I wonder why these 3 datasets were chosen, rather than other SST variants, both extending over the pre- and post-satellite periods, e.g. COBE, ERSST v5, GISTEMP, etc

Have the authors investigated the sensitivity of their analysis (and primary conclusions) to the input data choices?

Page 5, lines 87-89: SST coverage is relatively sparse prior to the near-global coverage of satellite

SST. Consequently, I wonder whether a step-like change in the data availability may have led to a step-like response? It appears that in Fig 4B, D, F, the largest changes correspond closely to the satellite transition 1982-onward. Has the influence of the "satellite step" been investigated?

Page 5, lines 96-97: The climatological period chosen (1983-2012) is already impacted by anthropogenic influence on SST increases, so selecting this clim period will lead to conservative estimates of change. You should point this out to a reader (and a reviewer).

Page 6, lines 108-109: How does this "average increase of 1.6 annual events" compare to the biological response – so linking the physical response quantified against reported biological response (e.g. coral bleaching events)

Page 6, lines 116-118: There may be a need to for caution when using satellite SST estimates as cloudiness can impact retrievals (e.g. Wentz et al., 2000; Trenberth & Fasullo, 2010)

Page 6, lines 122-123: You may as well point out to a reader that 1982-1998 is half of the 1982-2016 period – you have split your satellite-period time series in half

Page 6, lines 132-133: While reading this, I wondered how long would a MHW need to extend before biological impacts occur? I can imagine this would be ecosystem dependent, but would be an interesting background number(s) to present to a reader – are these changes biologically significant?

Page 7, lines 139-141: 1.3 days per decade longer, how does this relate to the earlier number of 5-10 for tropical regions (p6, lines 132-133), and is this number area-weighted? If yes, does it make much of a difference? I would assume that tropical regions run much closer to the upper limit in which biological impacts are "felt", so are a more naturally important region of analysis.

Page 10, lines 230-232: I seem to recall there are long records at Rottnest Island (~1960- Western Australia) and Maria Island (~1945- Tasmania). Are other non-Northern Hemisphere datasets, e.g. Great Barrier Reef that may also be considered?

Page 11, line 254: "0.3 to 1.5 annual events" Area-weighted? As it appears (Fig 4) larger positive anomalies are found in tropical regions, and as these regions represent a larger surface area than polar regions such statistics are biased to the tropics

Page 13, line 325: "maintaining long-term records.." A correspondence last year noted potential issues with maintenance of the global ocean observing system (e.g. Durack et al, 2016). It would be useful to highlight to a reader that the only way information on continuing changes can be quantified, is to maintain the observing system, both insitu and remote.

Page 18, lines 459-463: Which versions of HadISST and HadSST3 have been used? I also note that HadISST1 and HadISST2 SSTs benefit from satellite coverage from 1982 onward (see Section 3.6 in Rayner et al., 2003), so the statement "in situ observations have been interpolated onto a 1 x 1 horizontal grid (HadISST)" should be amended to reflect the correct information.

Figures and tables

Figure 1: For the global averages, are these numbers area-weighted? If yes, does it change the number much? The quality of the figures could also be improved, as hatching (panels B, E, H and K) was difficult to distinguish

Figure 2: The comment noted above (Page 5, lines 87-89), I wonder what would a full 1870-2016 version of Fig 2 look like?

Figure 3: It's curious to me that only 3 of the 15 panels show a statistically significant result. I wonder if data quality in the early part of the datasets is something that can be discussed? It would be useful to augment Supp S1.1 with any analysis that exists to validate these timeseries data.

Figure 5: the muted response in the MEI/PDO/AMO removed (black) lines suggests that variability is driving much of the large changes, not the anthropogenic forcing

References

- Banzon, V. et al (2016) <https://doi.org/10.5194/essd-8-165-2016>
Barriopedro, D. et al (2011) <https://doi.org/10.1126/science.1201224>
Durack, P. J. et al (2016) <https://doi.org/10.1038/nclimate2946>
Meehl, G. A. & C. Tebaldi (2004) <https://doi.org/10.1126/science.1098704>
McMichael, A. J. et al (2006) [https://doi.org/10.1016/S0140-6736\(06\)68079-3](https://doi.org/10.1016/S0140-6736(06)68079-3)
Mitchell, D. et al (2016) <https://doi.org/10.1088/1748-9326/11/7/0744006>
Poumadere et al. (2005) <https://doi.org/10.1111/j.1539-6924.2005.00694.x>
Trenberth, K. E. & J. T. Fasullo (2010) <https://doi.org/10.1175/2009JCLI3152.1>
Wentz, F. J. et al (2000) <https://doi.org/10.1126/science.288.5467.847>

Reviewer #4 (Remarks to the Author):

Overall I like this manuscript a lot, and believe it has the potential to make a valuable contribution to the current understanding for trends and variations in the history of marine heat wave events (MHWs) as documented by global, gridded SST data sets. It is a timely issue, with recent extended periods of extreme ocean warming leading to widespread and well-documented impacts on living marine resources. I do have a few issues I'd like the authors to consider before I can recommend this manuscript is accepted for publication, with the more important issues listed 1-4, and additional specific comments below.

1. I think their methods for removing the MEI from the various time series needs better explanation (see my comment below).
2. I think they need to account for the fact that the MEI and PDO index are not independent.
3. It is not clear to me that the 5 in situ records analyzed here offer much support for the key findings based on global gridded data sets. Specifically, the in situ records show less dramatic trends in MHW statistics from the 1920 to early 2000s, while the global gridded time series suggest both secular trends over that period of record and a substantial increase in the most extreme events in the period from 1982-2016.
4. I also believe that the extraordinary spike in MHW statistics in the 2014-2016 period shown in this work, which remain even after removing the linear-regressions onto the MEI, PDO, and AMO indices, deserve more discussion. While I don't want to ask the authors to take a stab at estimating the relative importance of natural versus forced climate change in an attribution study, I would like at least an estimate of the likelihood of such an event based on the historical record from 1950-2010.

In summary, I think there is a lot of interesting and important material in this manuscript and that it has the potential to make a significant contribution, but I would the author to make some mostly minor revisions.

line 40, 87: I am not a fan of using "Since the early 20th century" – please specify the start/end dates of analysis: between 1925-1954 and 1987-2016

line 84: please be more specific, Replace "long-term" with "century-long", "century-scale", "secular" or "linear" to better distinguish this from multi-decadal variations

line 153: Is the standard deviation of OISST calculated on daily gridded time series after the mean seasonal cycle is removed? Please clarify either here or in the Methods section.

line 156: Does your distinction between long-term climate change and natural climate variability rest on an implicit assumption that "long-term climate change" = radiative forcing/thermodynamic response, while "natural climate variability" = internal dynamical modes? It is not so clear to me that this distinction is warranted given the literature suggesting forced changes in decadal modes of variability (specifically PDO and AMO). I think you need to try to be more careful with what you want to say here. If you believe the ENSO, PDO, AMO variations are "all natural", state that assumption explicitly.

line 193: What is "SST" referring to here, trend, or decadal change?

lines 229-232: Do these sites add much value to the analysis? Aren't there other sites equally long and in other parts of the world? I guess one interesting point is that the characteristics of your MHW statistics at the 3 California sites are somewhat different, and that is interesting for site-level variations in a very small region that is known to have highly correlated interannual variability in SST related to large-scale patterns (PDO, ENSO, etc.)

lines 241-243: How did you develop MHW proxies using the five long-term in situ station-based records?

lines 262-263: need to revise this sentence, something is missing here.

lines 291-293: That the impact of natural modes of variability alone can be isolated and then removed from the data is your assumption and you do this using simple linear models, please make that more clear.

line 300: replace "back to 1900" with "from 1900 to 2016"

lines 318-322: I don't think it makes sense to say that "the impacts of MHWs are more difficult to detect ... unless major impacts are observed ... this is problematic because where they've been observed they have resulted in major impacts". It seems to me that major events have been observed because they have caused major ecological impacts. Those major impacts are obvious to people that are tuned into living marine resources, and the events are also easily observed with in situ and remote measurement systems.

For instance, the recent global coral bleaching and tropical marine heat wave of 2014-2016 was headline news around the world for months. The ecological impacts of the NE Pacific MHW of 2014-15 was also easily observed all along the west coast of North America.

line 395: The MEI is available as 2-month averages, so please explain in more detail how you regressed daily SST onto this index.

line 404: replace "is" by "it"

line 409-410: I am surprised that you could only find these 5 in situ daily SST records with continuous data for over 75 year periods. Have you looked at the daily BC Lighthouse temperature records archive by Canada's Department of Fisheries and Oceans? See:

<http://open.canada.ca/data/en/dataset/719955f2-bf8e-44f7-bc26-6bd623e82884>

line 413-414: Did you evaluate the Scripps Pier SST record for possible inhomogeneities in the daily SST variance pre and post 1988?

line 508-512: It is not likely valid to treat the MEI and PDO indices as independent predictors. One alternative for dealing with the ENSO-PDO collinearity is to remove the part of the PDO that is related to ENSO through a stochastic climate model ... could do the same for AMO (see Newman et al. 2003. ENSO-forced variability of the PDO. *J. Climate*, 16, 3853-3857.)

SI Table 1: Why are the 5 station-based records not updated to 2016? I think they should all be updated so that the analyses all end with 2016 data.

Response to Reviewers

We would like to thank all reviewers for their careful reading and constructive comments on this manuscript. We believe it is now much stronger after addressing their concerns. The reviewer's comments are listed point-by-point below with the reviewer's original comments in italics and our responses in bulleted roman text.

Reviewer #2

This study investigates changes in marine heat wave characteristics over the last century based on a multi-dataset approach using satellite and in-situ measurements of SST. The authors apply the marine heat wave framework developed by Hobday et al. and find a significant increase in global heat wave characteristics, modulated by strong interannual-to-decadal variability.

The authors address a very important and timely topic. The manuscript is very well written and I find no major flaws in the results and methods. Congratulations. However, I have a few comments that prevent me to accept the manuscript in this present form. I am happy to accept the manuscript if the authors can adequately address my concerns raised below.

Major points:

I think the title as it currently stands is too general and in fact it has not been demonstrated properly (using a proper attribution framework such as described by Stott et al. 2004) that (anthropogenic-induced) ocean warming is responsible for more frequent and longer marine heat waves. In addition, the title should indicate that the authors investigate changes over the satellite period and the past century and not future changes. This is not clear with the current title.

- We have changed the title to: “Longer and more frequent marine heatwaves over the past century”

What is the sensitivity of the results to the definition of heatwaves. The authors currently just apply one possible definition of heat waves (i.e. 90th percentile and at least five consecutive days). How would the results change when a higher percentile would be considered? I think this could be easily tested, quantified and discussed.

- Done.
- To address this comment, we have run the analysis on the satellite data using 90th, 95th and 98th percentiles to define the MHW threshold. The four MHW metrics considered (frequency, duration, intensity, total days) each maintain a statistically significant ($p < 0.05$) positive trend regardless of the percentile used. The choice of percentile does affect the mean and trend quantitatively, as expected since e.g. the 90th percentile leads to a maximum of 10% of the time in MHW conditions (although the actual % is smaller due to the 5 day minimum duration), while for the 98th percentile, this maximum drops to 2%. Nonetheless, our conclusions around significant long-term change are not sensitive to the choice of percentile.
- We have added the following to the Methods (Section 1): “Note that the statistical significance of the long-term trends presented in the main text (described in the methods below) are not sensitive to the choice of percentile threshold (90th, 95th, and 98th percentiles tested; not shown), and therefore our conclusions about changes in MHW properties are robust to the choice of threshold.”

Also, what is the motivation behind the criteria ‘for at least five consecutive days’? Is there any literature available that justifies this criteria for all marine species? In any case, what is the average duration of a heat wave when just the 90th percentile is applied? Isn’t it anyway longer than 5 days?

- We use the 5-day threshold based on the recommendation of Hobday et al. (2016). This definition in turn is based on atmospheric criteria for a heatwave e.g. Perkins and Alexander (2013): ‘On the measurement of heat waves’ uses a 3-day definition and this is mentioned in the Hobday et al. (2016) paper. Indeed, the majority of events are longer than 5 days but there do exist “heat spikes” on shorter time scales (1-4 days) that are excluded using this definition. However, since our focus is on the climatological characteristics and long-term change of marine heatwaves in general, and not on the species-specific impacts, we considered it beyond the scope of the present study to calculate trends given multiple minimum duration thresholds, and have used the published recommendation. We do note that there is ongoing experimental work to determine duration-specific thresholds for impacts on marine organisms. Once this work is completed it can inform the duration threshold for applications of marine heatwave studies on specific organisms.

Hobday, A. J., Alexander, L. V., Perkins, S. E., Smale, D. A., Straub, S. C., Oliver, E. C., ... & Holbrook, N. J. (2016). A hierarchical approach to defining marine heatwaves. *Progress in Oceanography*, 141, 227-238.

Perkins, S. E., & Alexander, L. V. (2013). On the measurement of heat waves. *Journal of Climate*, 26(13), 4500-4517.

Minor points:

L.84-92: This paragraph seems to summarize all results and reads like an abstract. It does not really fit here.

- This paragraph is in line with Nature Communications style. Specifically, it is a requirement (see Manuscript Checklist on page linked below) that the final paragraph of the introduction “contains a brief summary of both the results and conclusions (written in present tense)”.

<https://www.nature.com/ncomms/journal-policies/editorial-process>

l. 366-367: It is not clear to me how the climatological baseline from the 1983-2012 period was calculated. Did you detrend the data before calculating the 90th percentile?

- We did not remove the trend. This ensures the climatology is consistent with the data we are comparing it against, considering we wish to quantify the trends in MHW properties, consistent with existing practice in the analysis of atmospheric heatwaves (e.g. Perkins et al 202). We have now specified that we have analysed *raw* SST data, without further processing of any kind (trend removal, homogeneity adjustments, etc).

Perkins, S. E., Alexander, L. V., & Nairn, J. R. (2012). Increasing frequency, intensity and duration of observed global heatwaves and warm spells. *Geophysical Research Letters*, 39(20).

l.388-389. What do you mean with ‘moderately warming anomaly events’? Please clarify.

- This sentence has been revised as: “The MHW definition identified moderately warm anomaly events, meaning those of weaker intensities and without strong impact, in addition to the most extreme and impactful events.”

l.390-391: Yes, but if they are not rare events, they may not have an impact on organisms and ecosystems? In light of this, can you still justify your text in the introduction that states that especially extreme events may have significant impacts on marine ecosystems?

- The more extreme events would also be the more rare events. These two properties are strongly correlated and generally assumed to go together, and so we would expect these extreme and rare events to have impacts on marine ecosystems. In light of this, the speculation in the introduction (that the especially extreme events may have significant impacts on marine ecosystems) remains valid.

l. 326-328: I somehow disagree with this statement. I think a proper risk assessment for all marine species cannot be done with only one definition for marine heatwaves. The vulnerability of species to temperature changes is likely species dependent. Therefore, a unified framework will not help that much for a risk assessment.

- This statement was intended to focus on the physical aspects of MHWs, rather than their ecological implications. We agree with the reviewer that the ecological impacts of MHWs will be definition-dependent. We have therefore modified the sentence to clarify that we are referring to the physical properties of MHWs: “In addition, documenting events within a consistent framework across space and time will enable the comparison of the physical properties of different MHWs and contribute towards a greater understanding of their distribution and drivers.”

Reviewer #3

Many thanks for providing me with the opportunity to review the submission by Oliver and co-authors. I thoroughly enjoyed reading this new contribution, which discusses the topic of heatwaves in the marine environment. I must apologise to the Editor and the Authors for my tardy review.

The presented analysis provides a good overview for the reader, however I believe a number of key points and opportunities to link the present work to existing heat wave literature have been missed in the current version. I also think the text should more comprehensively cover existing literature, in which it appears considerable work has been already published. I also found some of the text wordy, and difficult to parse. The manuscript would benefit from a start to finish re-read to improve flow.

- We thank the reviewer for their suggestions. We have increased our cover of heatwave literature (see below) and revised the manuscript to generally improve the flow of the text.

I have raised a number of queries about the selection of datasets, and raise the question of whether dataset selection matters to the key conclusions of the present work. While the MetOffice (HadISST and HadSST) datasets do benefit from considerable quality control, due to data sparsity subjective decisions are made during dataset construction. The impact of these decisions is not always obvious, but can be evaluated when multiple observational products are compared and contrasted alongside one another.

- The reviewer raises an important point. We have now increased the number of global, monthly datasets to five and reduced errors by averaging results across datasets. We believe this has substantially improved the results pertaining to the MHW proxy records. See details below in our response to the reviewer's specific comments.

In addition to the general comments above, I have added a number of specific points below for consideration by the authors.

I believe that the present submission is relevant to a wide readership, and with further refinement should be published in the Nature Communications. Considering my comments above, and specific comments below, I suggest major revisions and look forward to reading an improved manuscript in press in the coming months.

Specific Comments

Page 3, lines 34-44: I would recommend a tightening up of the abstract, the contents of this paper are really an interesting read, yet the abstract didn't "sell" it to me

- Done.
- The abstract has been rewritten to more clearly highlight the key points of our study.

Page 4, lines 51-52: Don't bury the lede. Reverse this sentence, this happened, because, so.. "A redistribution of marine species, and reconfiguration of ecosystems has occurred due to pervasive increases to global SSTs..". There are a number of instances of reversed sentences throughout

- Done. Thank you, we reworded as suggested by the reviewer.

Page 4, lines 57-58: To point out to a reader that these extreme events AND their impacts really matter, I suggest augmenting to introduce some of the atmospheric heat wave literature (e.g. Meehl

& Tebaldi, 2004; Poumadere et al., 2004; McMichael et al., 2006; Barriopedro et al., 2011; Mitchell et al., 2016 for example and many others) and the very large mortality impact of these events. *Extremes events do matter! Now we're looking at the ocean...*

- Done.
- We thank the reviewer for suggesting this literature. We have included it in the introduction now as a motivation regarding the significance and impacts of extreme events.

Page 5, lines 79-83: Which versions of datasets were used?

- Done.
- We have added the version details, see below.

For OISSTv2, there is an AVHRR and an AVHRR+AMSR-E version of the data (extending from 2002-11), the Banzon et al (2016) citation is more up-to-date than Reynolds et al (2007).

- We have used the v2.0, AVHRR-only version of the data, which is now fully specified in the Methods (Section 2). We have now added the Banzon et al. (2016) citation when referring to the NOAA OISST V2 data.

For MetOffice products, the latest HadISST2 dataset is v2.2.0.0, and HadSST3 is v3.1.1.0.

- We have used HadISST1 v1.1 and HadSST3 v3.1.1.0, and this is now specified in the Methods (Sections 4.2 and 4.3). We could not use v2 of HadISST since the SST data are not available yet, only ice data are currently available*.

* <https://www.metoffice.gov.uk/hadobs/hadisst2/>

I wonder why these 3 datasets were chosen, rather than other SST variants, both extending over the pre- and post-satellite periods, e.g. COBE, ERSST v5, GISTEMP, etc. Have the authors investigated the sensitivity of their analysis (and primary conclusions) to the input data choices?

- Done.
- As mentioned above, the reviewer raises an important point. We have now included several additional gridded datasets which extend back at least to the early 20th century. In doing so, we have considered the analysis a multi-dataset approach and the results presented in the main text reflect the dataset mean. We feel these results are more robust than in the previous revision. The datasets we have included (in addition to the HadISST and HadSST3 datasets) are: NOAA Extended Reconstructed SST v5 (ERSST), Centennial in situ Observation-Based Estimates 2 SST (COBE), Coupled European Centre for Medium-Range Weather Forecasts (ECMWF) ReAnalysis 20C SST (CERA-20C) and Simple Ocean Data Assimilation v3 SST (SODA). Two of these added datasets (ERSST and COBE) are statistical analyses while the other two (CERA-20C and SODA) are dynamical analyses. We have not used GISTEMP as the ocean data are simply derived directly from ERSST. The main results presented in the manuscript are based on the five datasets with global coverage, i.e. HadISST, ERSST, COBE, CERA-20C and SODA, while the HadSST3 data are used only to explore the impact of (a) observational errors (Methods section S4.4) and (b) non-global coverage (Methods section S4.3), both of which are not possible to test with the five global datasets.
- This multi-dataset approach has modified all of the results and figures related to the MHW proxies (Figs. 4, 5; Supplementary Figs. 6, 7, 9-12) including a new supplementary figure

showing the MHW changes for the individual datasets (the figures in the main text shows the dataset-mean results).

Page 5, lines 87-89: SST coverage is relatively sparse prior to the near-global coverage of satellite SST. Consequently, I wonder whether a step-like change in the data availability may have led to a step-like response? It appears that in Fig 4B, D, F, the largest changes correspond closely to the satellite transition 1982-onward. Has the influence of the “satellite step” been investigated?

- This is a very good point, and one that has occurred to the authors as well. A true test of this hypothesis would require a long SST data set that only ingests *in situ* data, and this is available from HadSST3. Supplementary Fig. 12 compares the globally averaged time series from HadSST3 and the dataset-mean based on the five other products (which include the effect of satellite SSTs). The time series match well, including the pre- and post-1982 variability indicating that the presence or absence of satellite SSTs has little impact on the results.

Page 5, lines 96-97: The climatological period chosen (1983-2012) is already impacted by anthropogenic influence on SST increases, so selecting this clim period will lead to conservative estimates of change. You should point this out to a reader (and a reviewer).

- Done.
- This point is now made in the Supplementary Material, end of Section S1.3.

Page 6, lines 108-109: How does this “average increase of 1.6 annual events” compare to the biological response – so linking the physical response quantified against reported biological response (e.g. coral bleaching events)

- It is beyond the scope of the current paper to explicitly quantify the biological impacts of a particular increase in MHW metrics. We can however speculate, and the following has been added to the Discussion section: “It is evident from regional-scale studies that MHWs can cause widespread loss of habitat forming species such as kelps and corals, drive shifts in species distributions, alter the structure of communities and ecosystems, and have economic impacts on aquaculture and seafood industries through declines in important fishery species. [...] Such ecological impacts are likely to have become more prevalent with the intensification of MHWs over the last 100+ years.”

Page 6, lines 116-118: There may be a need to for caution when using satellite SST estimates as cloudiness can impact retrievals (e.g. Wentz et al., 2000; Trenberth & Fasullo, 2010)

- Done.
- This point has been added (with citations) to the Methods section describing the satellite data source (Methods Section 2).

Page 6, lines 122-123: You may as well point out to a reader that 1982-1998 is half of the 1982-2016 period – you have split your satellite-period time series in half

- Done.
- This point has been added to the previous paragraph, at the first introduction of these two “half” time periods.

Page 6, lines 132-133: While reading this, I wondered how long would a MHW need to extend before biological impacts occur? I can imagine this would be ecosystem dependent, but would be

an interesting background number(s) to present to a reader – are these changes biologically significant?

- Done.
- There is detailed experimental work currently in progress on this specific question. However, we can speculate based on existing studies and the following has been added to the Discussion section: “MHW impacts on marine species and ecosystems can occur on a range of timescales, with some species showing effects after a few days and others responding only after several months of elevated temperatures – and these impacts can last beyond the duration of the event itself (Wernberg et al. 2016) – responses that are also confounded by the thermal tolerance of different species living in the same region (Smale et al. 2015).”.

Smale, D. A., A. L. E. Yunnice, T. Vance and S. Widdicombe (2015). Disentangling the impacts of heat wave magnitude, duration and timing on the structure and diversity of sessile marine assemblages. PeerJ 3:e863; DOI 10.7717/peerj.7863.

Page 7, lines 139-141: 1.3 days per decade longer, how does this relate to the earlier number of 5-10 for tropical regions (p6, lines 132-133), and is this number area-weighted? If yes, does it make much of a difference? I would assume that tropical regions run much closer to the upper limit in which biological impacts are “felt”, so are a more naturally important region of analysis.

- The mean value of MHW duration in the tropics is 5-10 days. The global average increase in duration is 1.3 days/decade, which translates to an increase in average duration of ~4.5 days over the 35-year record, a large fraction of the mean value in the Tropics. However, the question “does it make a difference?” relates more to ecological impacts, which is beyond the scope of this paper and is part of ongoing work. We acknowledge your point: “I would assume that tropical regions run much closer to the upper limit in which biological impacts are “felt”, so are a more naturally important region of analysis”. However, it is notable that many mid-latitude regions have also been sites of major MHW impacts in recent years (northern Mediterranean in 2003, Western Australia in 2011, northwest Atlantic in 2012, southeastern Australia in 2015/16, northeast Pacific “blob” in 2014-16).
- Note that global-averages are area-weighted based on pixel area. This has been clarified in the Methods (Section 2).

Page 10, lines 230-232: I seem to recall there are long records at Rottnest Island (~1960- Western Australia) and Maria Island (~1945- Tasmania). Are other non-Northern Hemisphere datasets, e.g. Great Barrier Reef that may also be considered?

- We initially pursued the Australian National Reference Data stations (Rottnest Island, Maria Island, Port Hacking) as sources of long records of ocean temperature. However, the sampling rate at these sites (weekly-to-monthly) was too coarse and irregular to allow us to detect marine heatwaves with a comparable temporal resolution as is possible with the chosen station data. There are some sites along the Great Barrier Reef that have sub-daily data, but records only extend back to 1987, there are data gaps, and instruments have not necessarily been placed at the same exact depth or exact location – limiting the utility of such records for our study.

Page 11, line 254: “0.3 to 1.5 annual events” Area-weighted? As it appears (Fig 4) larger positive anomalies are found in tropical regions, and as these regions represent a larger surface area than polar regions such statistics are biases to the tropics

- This is not “area-weighted” as it is not a calculated statistic, but rather a visual description of Fig. 4A visually. The majority of the figure exhibits positive changes, in the range of 0.3 to 1.5 annual events locally, which is what was reported in the text.

Page 13, line 325: “maintaining long-term records..” A correspondence last year noted potential issues with maintenance of the global ocean observing system (e.g. Durack et al, 2016). It would be useful to highlight to a reader that the only way information on continuing changes can be quantified, is to maintain the observing system, both in situ and remote.

- Done.
- These points have been added to this part of the discussion, including a reference to Durack et al. (2016).

Page 18, lines 459-463: Which versions of HadISST and HadSST3 have been used? I also note that HadISST1 and HadISST2 SSTs benefit from satellite coverage from 1982 onward (see Section 3.6 in Rayner et al., 2003), so the statement “in situ observations have been interpolated onto a 1 x 1 horizontal grid (HadISST)” should be amended to reflect the correct information.

- Done.
- The dataset versions have been added (see related comment above), and we have now specified that satellite observations are also included in HadISST, from 1982 onwards.

Figures and tables

Figure 1: For the global averages, are these numbers area-weighted? If yes, does it change the number much?

- Global averages are area-weighted based on pixel area. This has been clarified in the Methods (Section 2). Nonetheless, the results are not strongly sensitive to area-weighting. See also our response above to the reviewer’s comment regarding *Page 7, lines 139-141*.

The quality of the figures could also be improved, as hatching (panels B, E, H and K) was difficult to distinguish

- Done.
- The figure hatching, and figure quality in general, has been improved throughout the manuscript.

Figure 2: The comment noted above (Page 5, lines 87-89), I wonder what would a full 1870-2016 version of Fig 2 look like?

- In early versions of this paper we considered having the start date in the mid-19th century. However, the data coverage globally is simply too sparse for such a figure to be reliable and defensible. Therefore, we omitted it. Nonetheless, if the reviewer is interested, we provide below a full 1871-2016 time series of globally-averaged MHW properties. Note that HadSST3 does not have any valid data points in the 19th century (based on our method of excluding years with <50% coverage globally).

Figure 3: It's curious to me that only 3 of the 15 panels show a statistically significant result. I wonder if data quality in the early part of the datasets is something that can be discussed? It would be useful to augment Supp S1.1 with any analysis that exists to validate these timeseries data.

- A majority of stations (4 of 6, see the additional station added in response to Reviewer #4) indicate a statistically significant increase in MHW frequency. The results are broadly consistent with our findings, which is that MHW frequency shows the strongest signal due to long-term warming.
- However, it should be noted that the long *in situ* station records are primarily included for the development of the global MHW proxy records and not for an analysis of the signals they provide. A deeper analysis of the station data is beyond the scope of the current study (but is being considered for future work). The primary reason for including the long station data is that they are needed for developing the proxies – they are the only data with daily values over such a long period which allow for testing the proxies (i.e. with long enough training and validation periods). The satellite data offer only a multi-decadal training period. Our simple analysis here has highlighted that a comprehensive analysis of all available stations, allowing for both long (~centennial) and relatively shorter (multi-decadal) records, and including a detailed analysis of homogeneity issues is certainly warranted.
- The following has been added to the beginning of the proxy development section of the Methods (Section 4), to make this point clear: “We developed proxies by selecting the MHW metrics we wished to predict (annual frequency, duration, intensity) and the set of variables which may be possible predictors (annual mean SST, annual maximum SST, annual count of months above a threshold, etc). Generalised Linear Models were trained on the long station series over the post-1982 period and validated over the pre-1982 period. This was used to select which variables should be used as predictors for the annual MHW metrics. Then these variables were used to train proxy models on the post-1982 MHW satellite data using predictors derived from the monthly SST analyses. Monthly SST

analyses have no daily data with which to validate the model selection – which is why the model selection was done on the long station time series, and why the use of the stations was important to the study.”

Figure 5: the muted response in the MEI/PDO/AMO removed (black) lines suggests that variability is driving much of the large changes, not the anthropogenic forcing

- Our findings indicate that the majority of the long-term warming – as measured by the difference in mean MHW metrics between 1925-1954 and 1987-2016 – is still captured by the time series after removing the modes (MEI/PDO/AMO). The change in MHW frequency (between 1925-1954 and 1987-2016) is reduced by only 13% after removing the effect of the three climate modes, and the change in MHW duration is in fact increased by 2% after removing the modes. The implication is that the majority of the warming is present regardless of these modes. The results of globally averaged changes for the three proxies, after the removal of the modes, has now been added to the manuscript along with text making the point that the removal of the modes in fact highlights the secular trend. It is true that the modes do contribute to a significant portion of the transient variability of the MHW metrics but it is this combination of variability (warm events) and long-term trend that likely impacts ecosystems, a point which has now been made in this paragraph discussing the results of Figure 5.

References

- Banzon, V. et al (2016) <https://doi.org/10.5194/essd-8-165-2016>*
Barriopedro, D. et al (2011) <https://doi.org/10.1126/science.1201224>
Durack, P. J. et al (2016) <https://doi.org/10.1038/nclimate2946>
Meehl, G. A. & C. Tebaldi (2004) <https://doi.org/10.1126/science.1098704>
McMichael, A. J. et al (2006) [https://doi.org/10.1016/S0140-6736\(06\)68079-3](https://doi.org/10.1016/S0140-6736(06)68079-3)
Mitchell, D. et al (2016) <https://doi.org/10.1088/1748-9326/11/7/0744006>
Poumadere et al. (2005) <https://doi.org/10.1111/j.1539-6924.2005.00694.x>
Trenberth, K. E. & J. T. Fasullo (2010) <https://doi.org/10.1175/2009JCLI3152.1>
Wentz, F. J. et al (2000) <https://doi.org/10.1126/science.288.5467.847>

Reviewer #4

Overall I like this manuscript a lot, and believe it has the potential to make a valuable contribution to the current understanding for trends and variations in the history of marine heat wave events (MHWs) as documented by global, gridded SST data sets. It is a timely issue, with recent extended periods of extreme ocean warming leading to widespread and well-documented impacts on living marine resources. I do have a few issues I'd like the authors to consider before I can recommend this manuscript is accepted for publication, with the more important issues listed 1-4, and additional specific comments below.

1. I think their methods for removing the MEI from the various time series needs better explanation (see my comment below).

- Done.
- We have addressed the reviewer's comments below and feel that this methods description is now clearer as a result.

2. I think they need to account for the fact that the MEI and PDO index are not independent.

- Done.
- The MEI is correlated with the PDO index at $r=0.32$ over the 1905-2016 period. To control for this dependence, we have now removed the component of the PDO index that is linearly related to the MEI, using linear regression. The results are not sensitive to this change. As an example, the figure below shows the global mean MHW time series (from HadISST proxies) after removing the influence of the PDO, for both the method presented in the original submission (black) and the revised method (red). It can be seen that this change has minimal impact on the results.

Neither the MEI nor the PDO index are significantly correlated with the AMO index, and so we do not consider there is a need to perform a similar correction to the AMO index.

- Text to this effect has now been added to the Methods (Section 4.3).

3. *It is not clear to me that the 5 in situ records analyzed here offer much support for the key findings based on global gridded data sets. Specifically, the in situ records show less dramatic trends in MHW statistics from the 1920 to early 2000s, while the global gridded time series suggest both secular trends over that period of record and a substantial increase in the most extreme events in the period from 1982-2016.*

- The long in situ records are critical for the development of the global MHW proxy records. This is because, unlike the global gridded SSTs, they have long daily data records allowing for a validation of the proxy models against MHW metrics calculated directly from the daily data. This is necessary for the model selection step. Please find more details in the responses below to comments related to lines 229-232 and 241-243.

4. *I also believe that the extraordinary spike in MHW statistics in the 2014-2016 period shown in this work, which remain even after removing the linear-regressions onto the MEI, PDO, and AMO indices, deserve more discussion. While I don't want to ask the authors to take a stab at estimating the relative importance of natural versus forced climate change in an attribution study, I would like at least an estimate of the likelihood of such an event based on the historical record from 1950-2010.*

- Done.
- The reviewer raises a very important point. To address this point, we have added the text below to the end of the results section (immediately after the results on the effect of removing the modes of variability): “Extraordinary global ocean warming occurred during the 2014-2016 period. Contributions to this warming included the 2014-2016 marine heatwave in the NE Pacific, the 2015/16 El Niño, the 2014 switch to a positive PDO phase and overall background warming (Su et al. 2017); this prolonged event was also more evident as it came after a period of global warming hiatus. This 2014-2016 warming was also evident in our globally averaged time series of MHW statistics (Fig. 5, blue lines). While the removal of the MEI and PDO influences reduced the magnitude of this warming (Fig. 5, black lines), consistent with the roles of these modes noted above, it was still evident. This was primarily due to the persistent marine heatwave in the NE Pacific (Bond et al. 2015) which was unprecedented in the historical record and cannot be fully explained by the coincidence of climate modes that occurred at the time (e.g. Oliver et al. 2017 for the related high-latitude Pacific warming of 2016).”

Su J., Zhang R. & Wang H. (2017) Consecutive record-breaking high temperatures marked the handover from hiatus to accelerated warming, *Scientific Reports* 7: 43735.

Oliver E.C.J., Perkins-Kirkpatrick S.E, Holbrook N.J. & Bindoff N.L. (2017) Anthropogenic influences on record 2016 marine heatwaves, accepted for publication in the *Bulletin of the American Meteorological Society's special supplement on Explaining Extremes of 2016*.

Bond N.A., Cronin M.F, Freeland H. & Mantua N. (2015) Causes and impacts of the 2014 warm anomaly in the NE Pacific. *Geophys. Res. Lett.* 42: 1-7

In summary, I think there is a lot of interesting and important material in this manuscript and that it has the potential to make a significant contribution, but I would like the author to make some mostly minor revisions.

line 40, 87: I am not a fan of using “Since the early 20th century” – please specify the start/end dates of analysis: between 1925-1954 and 1987-2016

- Done.

line 84: please be more specific, Replace “long-term” with “century-long”, “century-scale”, “secular” or “linear” to better distinguish this from multi-decadal variations

- Done.

line 153: Is the standard deviation of OISST calculated on daily gridded time series after the mean seasonal cycle is removed? Please clarify either here or in the Methods section.

- Done.
- The standard deviation was calculated from annual mean SSTs, and this is now clarified at this point in the text.

line 156: Does your distinction between long-term climate change and natural climate variability rest on an implicit assumption that “long-term climate change” = radiative forcing/thermodynamic response, while “natural climate variability” = internal dynamical modes? It is not so clear to me that this distinction is warranted given the literature suggesting forced changes in decadal modes of variability (specifically PDO and AMO). I think you need to try to be more careful with what you want to say here. If you believe the ENSO, PDO, AMO variations are “all natural”, state that assumption explicitly.

- Done.
- The reviewer makes an important point. We are not explicitly considering the modes of climate variability to be natural-only, although the language used in the manuscript was to that effect. We have added the following to the beginning of the subsection on “The role of internal variability”: “It is important when considering changes in MHWs to distinguish between the roles of secular climate change, which has in large part been attributed to anthropogenic factors, and of transient climate variability, which is largely intrinsic.”. We have furthermore removed references to “natural” variability and simply referred to “climate variability”, “climate modes”, etc.

line 193: What is “SST” referring to here, trend, or decadal change?

- We have clarified “maps” to “decadal change maps” in the text.

lines 229-232: Do these sites add much value to the analysis? Aren't there other sites equally long and in other parts of the world? I guess one interesting point is that the characteristics of your MHW statistics at the 3 California sites are somewhat different, and that is interesting for site-level variations in a very small region that is known to have highly correlated interannual variability in SST related to large-scale patterns (PDO, ENSO, etc.)

- We thank the reviewer for their insights into the long station records. However, a deeper analysis of the station data is beyond the scope of the current study (but is being considered for future work). The primary reason for including the long station data is that they are needed for developing the proxies (see response to the next comment)— they are the only data with daily values over such a long period which allows for testing the proxies (i.e. with long enough training and validation periods). The satellite data offer only a multi-decadal training period. Other sites (e.g. the Australian National Reference Data stations at Rottneest Island, Maria Island, Port Hacking or the Plymouth channel time series) have sampling rates (weekly-to-monthly) too coarse and irregular to allow us to detect marine heatwaves with a comparable temporal resolution as possible with the satellite data. There exist other daily records (WHOI, other California or UK sites) but are either too short (80 years is the

threshold used here) to suit the purposes of our present study (many start in the mid-20th century) or have large gaps in the data record (too many missing data). Note that on the reviewer's recommendation we have added one extra site from the British Columbia lighthouse dataset (see below). Our simple analysis here has highlighted that a comprehensive analysis of all available stations, allowing for both long (~centennial) and relatively shorter (multi-decadal) records, and including a detailed analysis of homogeneity issues is certainly warranted.

lines 241-243: How did you develop MHW proxies using the five long-term in situ station-based records?

- The following has been added to the beginning of the proxy development section of the Methods (Section 4) to clarify this question: “We developed proxies by selecting the MHW metrics we wished to predict (annual frequency, duration, intensity) and the set of variables which may be possible predictors (annual mean SST, annual maximum SST, annual count of months above a threshold, etc). Generalised Linear Models were trained on the long station series over the post-1982 period, and validated over the pre-1982 period. This was used to select which variables should be used as predictors for the annual MHW metrics. Then these variables were used to train proxy models on the post-1982 MHW satellite data using predictors derived from the monthly SST analyses. Monthly SST analyses have no daily data with which to validate the model selection – which is why the model selection was performed based on the long station time series, and why the use of the stations is important to the study.”

lines 262-263: need to revise this sentence, something is missing here.

- Done.
- We corrected the sentence, it now reads “Changes in the MHW duration proxy between the two periods showed an increase over 91% of the global ocean (Fig. 4C). The magnitude of the increase was typically up to 6 days but larger positive changes were found in the eastern tropical Pacific Ocean, northeastern Pacific Ocean, and parts of the South Pacific Ocean (6 to 14 days).”.

lines 291-293: That the impact of natural modes of variability alone can be isolated and then removed from the data is your assumption and you do this using simple linear models, please make that more clear.

- Done, this assumption has been added to the text.

line 300: replace “back to 1900” with “from 1900 to 2016”

- Done.

lines 318-322: I don't think it makes sense to say that “the impacts of MHWs are more difficult to detect ... unless major impacts are observed ... this is problematic because where they've been observed they have resulted in major impacts”. It seems to me that major events have been observed because they have caused major ecological impacts. Those major impacts are obvious to people that are tuned into living marine resources, and the events are also easily observed with in situ and remote measurement systems.

For instance, the recent global coral bleaching and tropical marine heat wave of 2014-2016 was headline news around the world for months. The ecological impacts of the NE Pacific MHW of 2014-15 was also easily observed all along the west coast of North America.

- This sentence has been revised to “Like the effects of atmospheric heatwaves on terrestrial ecosystems, which include widespread losses of crops and forests and reductions in biodiversity⁴⁸, the impacts of MHWs have resulted in major implications for societal interactions with the ocean – specifically in terms of fisheries, aquaculture, and tourism^{26,29,31}.”

line 395: The MEI is available as 2-month averages, so please explain in more detail how you regressed daily SST onto this index.

- Done.
- The following has been added to the text: “The MEI is defined monthly as a two-month average (Dec-Jan, Jan-Feb, etc) and we assumed the monthly values to be centred on the middle of the second month.” While this will implicitly impose a 15-day lag on the MEI with respect to the true “central date”, which should be the end of the 1st month / beginning of the 2nd month, we are not greatly concerned with how the regression is distributed across lags and we do not expect this 15-day difference to impact our +/-1 year lead-lag analysis.

line 404: replace “is” by “it”

- Done.

line 409-410: I am surprised that you could only find these 5 in situ daily SST records with continuous data for over 75 year periods. Have you looked at the daily BC Lighthouse temperature records archive by Canada’s Department of Fisheries and Oceans? See: <http://open.canada.ca/data/en/dataset/719955f2-bf8e-44f7-bc26-6bd623e82884>

- Done.
- The stations used are the only long (~centennial) series available with *daily* records (to our knowledge). We thank the reviewer for pointing us to the BC lighthouse data and we have now added this to our analysis. However, the only stations with records long enough (i.e. with start date of 1925 or earlier, as per the existing analyses) are Race Rocks (1921) and Departure Bay (1914). Departure Bay was missing too much data in the 1925-1954 period to make a useful time slice comparison, so we have used only data from Race Rocks.

line 413-414: Did you evaluate the Scripps Pier SST record for possible inhomogeneities in the daily SST variance pre and post 1988?

- No, we haven’t. Please see the response to “lines 229-232” above for a description as to the primary reason we have used the station data, and why a deeper analysis is beyond the scope of this study.

line 508-512: It is not likely valid to treat the MEI and PDO indices as independent predictors. One alternative for dealing with the ENSO-PDO collinearity is to remove the part of the PDO that is related to ENSO through a stochastic climate model ... could do the same for AMO (see Newman et al. 2003. ENSO-forced variability of the PDO. J. Climate, 16, 3853-3857.)

- Done.

- We have removed that part of the PDO index linearly related to the MEI (see response to major comment #2 above for details).

SI Table 1: Why are the 5 station-based records not updated to 2016? I think they should all be updated so that the analyses all end with 2016 data.

- Done.
- We have updated the station records to the most recent dates available, but this is not always 2016 as these data were not available for some stations. The end dates (rejecting partial years) are now 2013 (Newport Beach), 2014 (Scripps Pier, Pacific Grove, Port Erin) and 2016 (Arendal, Race Rocks). This update has meant we can use the same base period for both the stations analysis and the satellite data analysis (1983-2012) and we can use a more recent period for the time slice comparison (1984-2013).
- This has led to an update of the marine heatwave change results (Fig. 3 and associated text), the proxy development results (Supplementary Fig. 7 and associated text), and the large-scale correlation maps against the HadISST proxies (Supplementary Fig. 10). Regardless of these small quantitative changes to the results our conclusions remain unchanged.

REVIEWERS' COMMENTS:

Reviewer #2 (Remarks to the Author):

The authors have adequately addressed my comments and the points raised by the other reviewers. I particularly like the inclusion of the discussion of the sensitivity of the results to the choice of the percentile threshold. I am just wondering if there is any reason why the authors did not use the 99 percentile as usually used in other heat wave studies (why the 98P)?

I recommend the paper for publication after my very minor question has been addressed.

Reviewer #3 (Remarks to the Author):

Many thanks for providing me with the opportunity to review the revised submission by Oliver and co-authors. The revisions undertaken have considerably improved the manuscript, and it's a pleasure to see the review process has been beneficial for the authorship team and the evolution of this new and important contribution.

The addition of independent datasets to the analysis has considerably strengthened the key conclusions, as these revised results now express consistent changes for most of the available global SST datasets. I also believe that the expansion of the cited literature, which now includes atmospheric heat wave references, has considerably improved the manuscript – it's now clear to a reader (and this reviewer) that heat waves are starting to have a clear impact on society, both in the atmosphere and on land AND across the global ocean.

As a note to the Editor and authors, it is useful for a reviewer to have access to a tracked changes version of the revised manuscript. While I note the substantial changes, facilitated by a complete rewrite of many parts of the manuscript, being able to compare and contrast the old and new text is useful. It would also be helpful if these changes were more directly referenced in the response to reviewers, so key changes are easily located in the revised text.

I believe that this revised submission will be of interest to a wide readership, and should be published in Nature Communications. I look forward to reading this important manuscript in press in the coming months. Congratulations to the authorship team.

Reviewer #4 (Remarks to the Author):

I am pleased with the revisions made in response to reviewer comments, and recommend that this manuscript now be accepted for publication after only minor revisions. I congratulate the authors on writing a really interesting and valuable manuscript that should be appreciated by many readers.

Specific comments:

1. Supplemental Information: change "Frequency" to "Frequent" in the title
2. line 48: replace "since 1925" "from 1925 to 2016"
3. line 49: delete "have"
4. line 51: change "by the increase" to "by increases" because the former makes it sound like

there has been a uniform warming trend, but that is not the case

5. lines 124-127: I think you can delete this sentence. It doesn't support the previous sentence, and if you do keep it maybe you should move it after the next sentence for specific examples of the general statement about impacts on human health.

6. line 727 and 728: delete "has" before "increased"

7. line 730-731: change "the warming trend is accelerating" to "the warming trend accelerated over the 1925-2016 period"

8. end of line 802: change "and" to "an"

9. line 823: Change "Note that the statistical significance" to "The statistical significance"

10. line 999: delete "statistically analysed"

11. Line 1029: delete "Due to the sparse coverage ... for the remaining data," - no need to talk about "removed data before 1900" since the analysis doesn't start until 1900.

12. line 1091: delete repeated "are" here

13. lines 1143-1146. I'm trying to decipher your conceptual model here. My conceptual model includes internal versus external forcing, wherein the internal forcing includes basin-scale modes like ENSO, PDO, AMO, etc., as well as more local/regional forcing related to persistent atmospheric patterns like those that created the NE Pacific MHW in 2013/2014. External forcing can natural (e.g. volcanic origin aerosols, changes in the intensity of the sun) and anthropogenic radiative forcing (greenhouse gas and aerosol concentrations).

I would revise this sentence to "The variability of SST at a given location, region, or basin can be forced by local to large-scale climate variability, e.g. persistent atmospheric patterns or shifts in large-scale climate modes over century-long time scales under anthropogenic climate change."

14. References: Cavole et al. 2016 is listed twice, as reference 12 and 14

Response to Reviewers

We would like to thank all reviewers for their reading of and comments on this revised manuscript. The reviewer's comments are listed point-by-point below with the reviewer's original comments in italics and our responses in bulleted roman text.

Reviewer #2

The authors have adequately addressed my comments and the points raised by the other reviewers. I particularly like the inclusion of the discussion of the sensitivity of the results to the choice of the percentile threshold. I am just wondering if there is any reason why the authors did not use the 99 percentile as usually used in other heat wave studies (why the 98P)?

I recommend the paper for publication after my very minor question has been addressed.

- There was no specific reason why the 99th percentile was not used. The use of the 90th, 95th and 98th percentiles gives a good sampling of the upper tail of the distribution and we do not expect the results for the 99th percentile to be inconsistent with the results presented for the other high-percentile thresholds.

Reviewer #3

Many thanks for providing me with the opportunity to review the revised submission by Oliver and co-authors. The revisions undertaken have considerably improved the manuscript, and it's a pleasure to see the review process has been beneficial for the authorship team and the evolution of this new and important contribution.

The addition of independent datasets to the analysis has considerably strengthened the key conclusions, as these revised results now express consistent changes for most of the available global SST datasets. I also believe that the expansion of the cited literature, which now includes atmospheric heat wave references, has considerably improved the manuscript – it's now clear to a reader (and this reviewer) that heat waves are starting to have a clear impact on society, both in the atmosphere and on land AND across the global ocean.

- We thank the reviewer for their previous suggestions and are happy to see the revisions have satisfied their concerns.

As a note to the Editor and authors, it is useful for a reviewer to have access to a tracked changes version of the revised manuscript. While I note the substantial changes, facilitated by a complete rewrite of many parts of the manuscript, being able to compare and contrast the old and new text is useful. It would also be helpful if these changes were more directly referenced in the response to reviewers, so key changes are easily located in the revised text.

- We agree completely with the reviewer's sentiment. In fact, with our revised manuscript we supplied both a track-changed version and a clean version (all changes accepted) of the manuscript. It appears the track-changed version was not made available to the reviewer, which is unfortunate.

Reviewer #4

I am pleased with the revisions made in response to reviewer comments, and recommend that this manuscript now be accepted for publication after only minor revisions. I congratulate the authors

on writing a really interesting and valuable manuscript that should be appreciated by many readers.

Specific comments:

1. Supplemental Information: change “Frequency” to “Frequent” in the title

- Done.

2. line 48: replace “since 1925” “from 1925 to 2016”

- Done.

3. line 49: delete “have”

- Done.

4. line 51: change “by the increase” to “by increases” because the former makes it sound like there has been a uniform warming trend, but that is not the case

- Done.

5. lines 124-127: I think you can delete this sentence. It doesn’t support the previous sentence, and if you do keep it maybe you should move it after the next sentence for specific examples of the general statement about impacts on human health.

- Done. As suggested, we have moved this sentence after the statement about human health impacts.

6. line 727 and 728: delete “has” before “increased”

- Done.

7. line 730-731: change “the warming trend is accelerating” to “the warming trend accelerated over the 1925-2016 period”

- Done.

8. end of line 802: change “and” to “an”

- Done.

9. line 823: Change “Note that the statistical significance” to “The statistical significance”

- Done.

10. line 999: delete “statistically analysed”

- Done.

11. Line 1029: delete “Due to the sparse coverage ... for the remaining data,” - no need to talk about “removed data before 1900” since the analysis doesn’t start until 1900.

- This sentence was there to justify why we started the analysis in 1900. We have therefore left it in, but changed “removed data before 1900” to “restricted the analysis period to data since 1900”.

12. line 1091: delete repeated “are” here

- Done.

13. lines 1143-1146. *I’m trying to decipher your conceptual model here. My conceptual model includes internal versus external forcing, wherein the internal forcing includes basin-scale modes like ENSO, PDO, AMO, etc., as well as more local/regional forcing related to persistent atmospheric patterns like those that created the NE Pacific MHW in 2013/2014. External forcing can natural (e.g. volcanic origin aerosols, changes in the intensity of the sun) and anthropogenic radiative forcing (greenhouse gas and aerosol concentrations).*

I would revise this sentence to “The variability of SST at a given location, region, or basin can be forced by local to large-scale climate variability, e.g. persistent atmospheric patterns or shifts in large-scale climate modes over century-long time scales under anthropogenic climate change.”

- Done. We have replaced this sentence with the reviewer’s suggestion.

14. References: Cavole et al. 2016 is listed twice, as reference 12 and 14

- Fixed.